# Feynman-Kac Operator Expectation Estimator: An Innovative Method for Enhancing MCMC Efficiency and Reducing Variance

## Abstract

The Feynman-Kac Operator Expectation Estimator (FKEE) is an innovative method for estimating the target Mathematical Expectation $\mathbb{E}_{X \sim P}[f(X)]$ without relying on a large number of samples, in contrast to the commonly used Markov chain Monte Carlo (MCMC) algorithm. This method uses Physically Informed Neural Networks (PINN) to approximate the Feynman-Kac operator. It enables the incorporation of existing diffusion bridge models into the expectation estimator, and significantly improves the efficiency of using Markov chains while substantially reduces the variance. Additionally, this method mitigates the adverse impact of the curse of dimensionality, weakening the assumptions on the distribution of $X$ and $f$ in the general MCMC expectation estimator. In the algorithm implementation, the first step involves constructing a diffusion bridge over the target distribution or known data by matching the coefficients of the diffusion bridge from the random flow trajectories or a Markov chain. Subsequently, we employ PINN to solve the Feynman-Kac equation, and the solution of this equation provides the mathematical expectation in analytical form. Finally, we demonstrate the advantages and potential applications of this method through various concrete experiments, including the challenging task of approximating the partition function in the random graph model such as the Ising model.

## 1 Introduction

Markov Chain Monte Carlo (MCMC) is a widely utilized statistical computational method in various fields, such as statistics, machine learning, and computational science. It is mainly used in sampling from complex distributions, Bayesian inference, optimization etc. (Hesterberg, 2002; Ahmed, 2008). As for the purpose of MCMC algorithms, they can be divided into two categories. One category involves sampling from the target distribution. In recent research, about sampling from the target distribution, there has been exploration of alternative approaches to traditional MCMC samplers, such as generative models (Adler & Lunz, 2018) and diffusion models (Ho et al., 2019). The second category focuses on estimating the statistical characteristics of the target distribution, such as the expectation. MCMC is often applied to estimate the target Mathematical expectation and there are no widely accepted alternatives to the MCMC expectation estimator so far, except some special algorithms which are designed to estimating the Mathematical expectation $\mathbb{E}_{X \sim P}[f(X)]$ for some special probability distributions $P$ and performance functions $f$ (Tokdar & Kass, 2010). Since estimating the expectation often plays a key role in many application scenarios, it is very important to propose a suitable MCMC expectation estimator. A natural thought is to combine modern sampling approaches with the MCMC expectation estimator.

**Advantages and disadvantages of MCMC**: The effectiveness of the MCMC algorithm in sampling from a target distribution can be attributed to its utilization of empirical measures derived from samples at the terminal time. This approach leverages the law of large numbers for estimating mathematical expectations. Furthermore, it involves averaging over the values along the path, in accordance with the ergodicity theorem of the Markov chain. Notably, this methodology proves particularly advantageous in dealing with high-dimensional distributions, mitigating the challenges posed by the curse of dimensionality in integral approximations.

Nonetheless, a drawback of conventional MCMC algorithms is the prolonged burn-in period often required for the target distribution to reach a state of stationarity within the Markov chain. Additionally, to achieve accurate estimates of mathematical expectations, a substantial number of sample points $N$ is necessary, leading to estimation variances on the order of $\mathcal{O}(\sqrt{N})$. Using the Quasi-Monte Carlo method (Caflisch, 1998) the order of the variance is $\mathcal{O}(N^{\frac{1}{2}+\delta})$ where $\delta \leq \frac{1}{2}$. This not only diminishes the efficiency of the MCMC algorithm but also introduces bias. Estimation of error probabilities can be facilitated through concentration inequalities (Lugosi, 2003), which may depend on the Lipschitzian norm of $f$. Therefore, the careful selection of appropriate proposal distributions in MCMC algorithms becomes critical for achieving efficient and precise estimations of mathematical expectations. Another limitation of traditional MCMC algorithms surfaces when handling discrete random variables and complex functions $f$, resulting in high variance. Consequently, obtaining accurate estimates often necessitates a large quantity of points in the Markov chain, particularly pronounced in larger models. This scenario frequently occurs in random graph models (Cipra, 1987; Newman et al., 2002; Drobyshevskiy & Turdakov, 2019).

To enhance the MCMC algorithm, we propose the following improvements. The transition density function associated with the discrete Markov chain generated by the MCMC algorithm can be interpreted as the transition density function of a specific stochastic differential equation (SDE) of Markovian properties. In this study, we refer to this SDE as the **Diffusion Bridge model**. This encompasses a broad class of SDEs that share identical transition densities with the Markov chains in the MCMC algorithm. The distribution of the terminals in such SDEs aligns with a predefined target distribution, which can take the form of discrete points or a probability density function. Moreover, the starting point of this SDE can be either arbitrary or fixed.

Our rationale for this approach lies in the fact that a substantial number of burn-in samples go to waste when estimating Mathematical expectations by using the MCMC algorithm. However, these samples harbor valuable information, specifically pertaining to the gradient information of the drift and diffusion coefficients along the paths derived from the SDE. We capitalize on this information by integrating it through the Physics-Informed Neural Network (PINN) approach (Sharma & Shankar, 2022; Raissi et al., 2019; Yuan et al., 2022). This process, akin to approximating the Feynman-Kac Operator, is referred to as solving the **Feynman-Kac model**. Notably, this approximation is meshless and effectively overcomes the curse of dimensionality. By amalgamating different combinations of the aforementioned models, we derive the Feynman-Kac Operator Expectation Estimator (FKEE).

**Our contributions can be summarized as follows:**

- **Establishing a Link Between Sampling Methods and High-Dimensional Partial Differential Equations:** We establish a connection between widely used sampling techniques and the intricate realm of high-dimensional partial differential equations. Our innovative approach not only introduces a fundamental algorithm but also explores the synergy of merging two prominent algorithmic classes. We substantiate this through experimental analysis, demonstrating the potential of our approach as a bridge between these two disparate domains.

- **Introducing a More Versatile Diffusion Bridge Model:** We introduce a highly adaptable diffusion bridge model. This model not only allows for the specification of target distributions at terminal moments but also facilitates the reconstruction of the entire Markov chain. It can be employed in conjunction with the Feynman-Kac model for expectation estimation, as well as independently for resampling target distributions to estimate expectations.

- **Wide Applicability Across Diverse Domains:** Our model finds utility across various domains, including variational inference, distribution resampling, and the substitution of loss functions represented as expectations. Its versatility empowers the model to learn more robust features with reduced variance and fewer underlying assumptions. Importantly, this method can estimate expectations without succumbing to the limitations imposed by the curse of dimensionality.

- **Enhanced Efficiency and One-Time Training:** Our method often exhibits superior efficiency. For a given class of distributions, a single training session is sufficient to derive the diffusion bridge coefficients. These coefficients can be stored as expectations for estimating

a range of distributions, and computation can be expedited by harnessing the capabilities of GPU acceleration.

- **Broadening the Horizons of MCMC Algorithms:** We make significant enhancements to the scope of the MCMC algorithm. Our approach leads to a more efficient utilization of Markov chains. Notably, it necessitates fewer assumptions since it does not rely on the law of large numbers and the Markov ergodic theorem. This approach holds promise as a means to approximate mathematical expectations when dealing with non-independent samples.

## 2 RELATED WORK

The diffusion model belongs to a class of stochastic differential equations, which are used to approximate the target distribution. It has been widely used for generative models (Dhariwal & Nichol, 2021), variational inference (Geffner & Domke, 2021; Kingma et al., 2021), etc. The diffusion bridge model is a variant of the diffusion model. Early development of diffusion bridge models involved simulating processes originating from two endpoints (Beskos et al., 2008). Alternative approaches for constructing diffusion bridge models are outlined by (Liu et al., 2022; Bladt & Sørensen, 2010). Since the diffusion bridge model essentially functions as a sampling algorithm, it plays a pivotal role in addressing the crucial task of high-dimensional distribution sampling. Sampling high-dimensional distributions is a fundamental task with applications across various fields. Common methods include MCMC, random flow, and generative models. Recent work includes stream-based methods (Müller et al., 2018; Yang et al., 2017; Matsubara et al., 2020; Strathmann et al., 2015; Tran et al., 2019), MCMC-based methods (Deng et al., 2020; Chen et al., 2014; Jacob et al., 2017) and generative models (Nichol & Dhariwal, 2021), score-based models (Song et al., 2020). Normalizing Flows (Albergo & Vanden-Eijnden, 2022).

These models can be broadly categorized into two groups: those based on given discrete points and those relying on a given density function. The former primarily serves for learning and generating real-world data such as text and images, while the latter is used for sampling, statistical estimation, and similar purposes. Notably, Langevin diffusion (Cheng et al., 2018; Xifara et al., 2013; García-Portugués et al., 2017) is a classical model within the latter category.

The Feynman-Kac model is a technique employed to solve partial differential equations (PDEs) by using deep learning. Deep learning has found application in solving PDEs of the Feynman-Kac equation type, as demonstrated by (Berner et al., 2020; Blechschmidt & Ernst, 2021). (Liang & Borovkov, 2023) highlights the approximation of Feynman-Kac type expectations through the approximation of discrete Markov chains, thereby enhancing the order of convergence. When employing PINN to solve Feynman-Kac type PDEs, the sampling algorithm can be linked to the path of the SDE. This approach enables the acquisition of adaptive sampling points from the paths of SDE, which proves more efficient than uniform point selection (Chen et al., 2023). Further analysis of the approximation error for this class of equations is presented in (De Ryck & Mishra, 2022).

## 3 NOTATIONS

$\mathbb{P}(X_t)$ denotes the distribution of $X_t$ and $\hat{\mathbb{P}}(X_t)$ denotes the empirical distribution of $X_t$. $diag(A)$ denotes the diagonal matrix of matrix $A$. $diag(A)_i$ denotes the i-th element on the main diagonal of the diagonal matrix of $A$. $C^2$ denotes the space of continuous functions with second order derivatives. $A_{i,j}$ denotes the elements of row $i$ and column $j$ of the matrix $A$.

## 4 MAIN METHODOLOGIES AND CONTRIBUTIONS

### 4.1 DIFFUSION BRIDGE MODEL

The sampling methods mentioned in related work can be generalized into a common framework: for most MCMC sampling methods, we can consider using a Markov-type SDE as follows:

$$dX_t = \mu(X_t, t)dt + \sigma(X_t, t)dW_t, X_0 = x_0$$

where $\mu : \mathbb{R}^d \times [0, T] \to \mathbb{R}^d$ is a vector-valued function , and $\sigma : \mathbb{R}^d \times [0, T] \to \mathbb{R}^{d \times d}$ is a matrix-valued function. $\{W_t\}_{t \geq 0}$ is a Brownian motion taking values on $\mathbb{R}^d$. For a non-stationary diffusion

model, these coefficients $\mu$ and $\sigma$ are assumed to satisfy some regularity conditions to ensure the existence and uniqueness of the strong solution (Särkkä & Solin, 2019). For diffusion models with stationary distribution, the uniqueness of stationary distribution should be satisfied.

For a given distribution $P$ or some discrete points of $P$, we need to encode the information of $P$ into $(X_0, \mu, \sigma, T)$. This encoding should ensure that the distance between $\mathbb{P}(X_T)$ and $P$ is sufficiently small. The loss of the encoding should be minimized, which is a common objective in many generative models. For convenience, we call this SDE the **diffusion bridge model**.

The encoding loss is from two components: the first component is the structural loss, typically induced by the accuracy of $(\mu, \sigma)$, and the second is the discretization loss, usually stemming from the need for a sufficiently large $T$ and the numerical discretization of the SDE. However, the encoding loss depending on the presence of a density form for the distribution $P$. Specifically, if we have the density of $P$, we can employ methods without structural loss, such as Langevin MCMC. In this case, the structural loss is zero, but there exists discretization loss. MCMC samplers based on transition density typically fall into this category as well. If we lack the density of $P$, the constructed SDE will simultaneously exhibit both types of losses. The core focus of the diffusion model is how to minimize these two losses. The error in $(\mu, \sigma)$ is controlled by a specific loss function, while the discretization loss is typically controlled by minimizing $T$ as much as possible and using a high-precision SDE solver.

We propose a diffusion bridge model to minimise encoding loss, formally, This method similar to the Neural SDE (Tzen & Raginsky, 2019; Kidger et al., 2021). Explicitly, we consider using the following Neural SDE :

$$dX_t = \mu_{\theta_1}(X_t, t)dt + \sigma_{\theta_2}(X_t, t)dW_t, X_{0,\theta_3} = x_{0,\theta_3}.$$

Where $(X_{0,\theta_3}, \mu_{\theta_1}, \sigma_{\theta_2})$ is a neural network that conforms to the corresponding dimensions, and we use an MLP with the activation function tanh for this network. Here the time $T$ and the time step size $h$ are parameters given in advance for the SDE solver. Diffusion bridge model matching means that we use neural network methods to find the appropriate $(X_0^*, \mu^*, \sigma^*)$ such that the distribution of $X_T$ at the moment $T$ is just the given target distribution $P$. We need to categorise the target distribution to determine the matching method. This depends on whether the target distribution has an explicit probability density function.

**For the case where there are only a few discrete observations :** We propose a matching algorithm that deals with only a subset of discrete points from the target distribution $P$. Specifically, we employ a diffusion bridge model to parameterize $(X_{0,\theta}, \mu_\theta, \sigma_\theta)$ using a neural network. Given the empirical distribution of the target $\hat{\mathbb{P}}$, we simulate $N$ trajectories of Brownian motion and use the Euler-Maruyama method to obtain $X_T$. Subsequently, we match the obtained solutions to the given points and utilize the Wasserstein distance loss function.

$$X_0^*, \mu^*, \sigma^* = \underset{(X_{0,\theta_3}, \mu_{\theta_1}, \sigma_{\theta_2})}{\arg\min} W_p(\hat{\mathbb{P}}(X_T), \hat{\mathbb{P}}).$$

Where $W_p(\hat{\mathbb{P}}(X_T), \hat{\mathbb{P}})$ is the computation of the Wasserstein distance for two empirical distributions. Since $T$ and $h$ are given, we can estimate the discretization loss and control the structural loss through the Wasserstein distance loss. There are two other uses for this method, as follows:

**For one use : Resampling samples.** For a given set of high-quality samples (Not within the burn-in period of MCMC.), we can also consider this method for resampling samples. This means using the given points to match a diffusion bridge model, and then simulating SDE to obtain more samples. Some of the high-quality samples can also be obtained by other sampling methods, such as Perfect Sampling (Djurić et al., 2002), Adaptive Metropolis, Importance Sampling (Haario et al., 2001), Differential Evolution Markov Chain Monte Carlo (DE-MCMC) (Fan, 2012),etc (Kim, 2000).

**For another use : Matching Markov chains and generating more Markov chains quickly.** The paths $Y_i^N$ are trajectories of N independently run Markov chains obtained from MCMC algorithms, where $i \leq M$. The objective is to determine a set of $(X_0, \mu, \sigma)$ such that, at $M$ moments, $X_i$ and $Y_i$ are close enough to each other in the sense of the Wasserstein distance. Here, $X_i$ is the solution to the SDE defined by $(X_0, \mu, \sigma)$. Specifically, we can achieve this by optimising the Wasserstein distance loss like follows:

$$X_0^*, \mu^*, \sigma^* = \underset{(X_{0,\theta_3}, \mu_{\theta_1}, \sigma_{\theta_2})}{\arg\min} \sum_{i=1}^{M} W_p(\hat{\mathbb{P}}(X_i), \hat{\mathbb{P}}(Y_i)))$$

In this matching process, we have the flexibility to match either a segment or the entire Markov chain. If we opt to match only a segment, we can strategically begin the matching process from a later moment, thereby minimizing reliance on points within the burn-in period.

**For the case where we know the target distribution :** This scenario has already been studied by MCMC algorithms and SDE-type samplers. In this case, we can still use our method to match a diffusion bridge and have two matching methods. The first one involves specifying a density function and then using existing MCMC algorithms to obtain $N$ discrete points at each position $X_t$. We then employ the same Wasserstein distance loss as mentioned above for matching. The second method involves using transition density matching. Specifically, given a density function $f$, we can determine a transition density function $p(y|x, h)$ in MCMC algorithms. Then, by discretizing the SDE using the Euler-Maruyama method, we obtain the following transition density:

$$\hat{p}(y|x, h) = \mathcal{N}(y; x + \mu_{\theta_1}(x, t)h, h\sigma_{\theta_2}(x, t)\sigma_{\theta_2}^T(x, t))$$

We can consider the following loss function :

$$X_0^*, \mu^*, \sigma^* = \underset{(X_{0,\theta_3}, \mu_{\theta_1}, \sigma_{\theta_2})}{\arg\min} \iint [\hat{p}(y|x, h) - p(y|x, h)]^2 dy dx + [X_0 - X_{0,\theta_3}]^2$$

**The design of the loss function here is not unique, The diffusion bridge matching method in our paper is just a baseline matching algorithm that can be replaced by many other algorithms.** But our method has the following advantages:

We consider using Neural SDE bridge for the following reasons: we need to minimize the number of steps to the target distribution in the smallest possible time interval due to the need to reduce the amount of PINN training. In cases with only partially observed samples, where the density information is absent, the matching process is not unique and relies on the chosen model. We use simulations of the same number of Brownian motion paths and use a fully trainable initial value of drift and diffusion coefficients to guarantee maximum flexibility, while the Wasserstein distance guarantees stability of the training The Wasserstein distance between the generated samples and the target value is also an overall match. Importantly, this approach does not involve the estimation of likelihood functions and maintains high efficiency. In summary, our matching model is specially designed for PINN training, offering the utmost flexibility and efficiency.training and retains maximum flexibility.

Directly using an SDE-type sampler as the diffusion bridge is also feasible, eliminating the need for a matching process. One of the most accessible methods for this purpose is the Langevin diffusion. This method is based on a standard diffusion equation that possesses a well-defined stationary distribution, making it a widely employed choice for sampling purposes. The set of parameter pairs that determine the diffusion bridge are $(X_0, \mu, \sigma, T)$. In this case, many of the algorithms for MCMC can be reduced to a diffusion bridge model in Table 1.

## 4.2 Feynman-Kac model

This section constitutes our primary contribution, as we present a novel perspective on understanding expectation estimation: The goal is to estimate $\mathbb{E}_{X \sim P}[f(X)]$, representing the decoding of $P$ to derive a deterministic value $\mathbb{E}_{X \sim P}[f(X)]$. All the information about $P$ is encoded within $(X_0, \mu, \sigma, T)$. The decoding loss determines the accuracy of our estimate for $\mathbb{E}_{X \sim P}[f(X)]$, and the decoding speed influences the algorithm's efficiency.

The decoding loss of classical MCMC expectation estimators might not be optimal because these estimators often do not directly incorporate information about $(X_0, \mu, \sigma, T)$. The reason is that we only need to simulate a subset of samples for averaging, which is often locally biased and lacks a comprehensive estimate for the entire distribution. The method of control functions (Oates et al., 2014; South et al., 2020) can be considered a post-processing approach and reuses information from $P$, but is not universally applicable to any $P$ and $f$. The need for stronger assumptions to guarantee accurate estimates, influenced by the law of large numbers and the ergodic theorem, further contributes to the diminished efficiency of MCMC algorithms. Therefore, the classical MCMC expectation estimator is an incomplete decoding.

Our key innovation lies in the direct utilization of information within $(X_0, \mu, \sigma, T)$, as it encapsulates all the information about $P$. This constitutes the essence of our algorithm. The decoding process

Table 1: Comparison of Diffusion bridge model

| Method | $X_0$ | $\mu$ | $\sigma$ | $T$ | Descriptions |
|---|---|---|---|---|---|
| Classical MCMC | $\forall \mathbf{x}_0 \in \mathbb{R}^d$ | $p(y|x)$ | $p(y|x)$ | $\infty$ | $p(y|x)$ is the transfer probability density function in the MCMC algorithm. The meaning of $p(y|x)$ is that the corresponding coefficients can be obtained by a SDE. |
| Langevin MCMC | $\forall \mathbf{x}_0 \in \mathbb{R}^d$ or $\forall X_0 \sim P_0$ | $\frac{1}{2}\nabla_x \log p(x)$ | $I_{d \times d}$ | $\infty$ | $p(x)$ is target density function and density function of a stationary distribution. $P_0$ is the initial distribution. |
| Score-based SDE and diffusion models (DDPM) | $\forall X_0 \sim P_0$ | $f(\mathbf{x},t) - g^2(t) \nabla_{\mathbf{x}} \log p_t(\mathbf{x})$ | $g(t)$ | $< \infty$ | $\nabla_{\mathbf{x}} \log p_t(\mathbf{x})$ is obtained from the data and $f(x,t)$ and $g(t)$ are known. $P_0$ is the prior distribution. |
| Flow match ODE | $\forall X_0 \sim P_0$ | $v(x,t)$ | $0$ | $1$ | $v(x,t)$ is obtained by matching the data. $P_0$ is the initial distribution. |
| Neural SDE bridge **(taken in this paper)** | $x_0 = x_{0,\theta_3}$ | $\mu_{\theta_1}(x,t)$ | $\sigma_{\theta_2}(x,t)$ | $< \infty$ | $\mu_{\theta_1}(x,t)$ and $\sigma_{\theta_2}(x,t)$ is obtained from the data or match method. |

can be seen as an approximation of the Feynman-Kac operator, and it can be formally obtained by solving the Feynman-Kac equation. In a formal manner, we introduce the Feynman-Kac operator as follows:

Feynman-Kac operator (Del Moral & Del Moral, 2004) is a key component in the mathematical framework that allows the translation between deterministic PDEs and stochastic processes through the Feynman-Kac formulae (Feynman-Kac equation). Feynman-Kac equation (Pham, 2014) is a powerful method for solving partial differential equations (PDEs) and related problems by linking them to stochastic processes. The basic idea is to represent the solution of a PDE as an expectation of a function of a stochastic process, and to use Monte Carlo methods to approximate this expectation. We can reverse the process to get the new methods to get MCMC results. In other words, **we can use the solution of PDE to obtain an accurate expression of the corresponding MCMC results**. In the $d$-dimensional case, we consider the Feynman-Kac equation on the interval $[0, T]$ :

$$\frac{\partial u(x,t)}{\partial t} + \sum_{i=1}^{d} \frac{\partial u(x,t)}{\partial x_i} \mu_i(x,t) + \frac{1}{2} \sum_{i=1}^{d} \sum_{j=1}^{d} \frac{\partial^2 u(x,t)}{\partial x_i \partial x_j} (\sigma(x,t)\sigma(x,t)^T)_{i,j} = 0$$

$$u(x,T) = f(x).$$

The solution to the Feynman-Kac equation at the initial time is $u(x_0, 0) = \mathbb{E}[f(X_T)|X_0 = x_0]$. We compute the same conditional expectation. The proof of this theorem involves using the Itô formula and properties of martingales with stochastic integrals. We will provide a brief explanation in the appendix. For more details, please refer to (Särkkä & Solin, 2019).

Calculating this equation involves computing the Hessian matrix of a function and some partial derivatives, which can be obtained by using any library with automatic differentiation such as `Pytorch`. In particular, If we only consider that the diagonal diffusion coefficients are $\sigma$, it can accelerate the algorithm. In training process, PINN can use the Neural Tangent Kernel (NTK) (Saadat et al., 2022) to analyze the training behavior. Such as in Langevin diffusion where $\sigma = I_{d \times d}$, we need to calculate the second-order partial derivatives of the main diagonal. In the usual case, we only need to calculate $f(x) = x$. For Neural SDE bridge, in order to reduce the amount of computation, we can only consider a diagonal diffusion matrix function $\sigma : \mathbb{R}^d \times [0, T] \to \Lambda(\mathbb{R}^d)$, where $\Lambda(\mathbb{R}^d)$ is the set of real-valued diagonal matrices. we can only calculate the second-order derivatives of the diagonal in function, to avoid is the whole diffusion matrix function. To achieve this goal, we can specifically design the following loss functions:

$$\mathcal{L}_1 = \iint_{\mathcal{D}\times[0,T]} \left[ \frac{\partial u_\theta(x,t)}{\partial t} + \sum_{i=1}^d \frac{\partial u_\theta(x,t)}{\partial x_i}\mu_i(x_t,t) + \frac{1}{2}\sum_{i=1}^d \frac{\partial^2 u_\theta(x,t)}{\partial x_i^2}diag(\sigma^2(x,t))_i \right]^2 dxdt$$

$$\mathcal{L}_2 = \int_{\mathcal{D}} \left[ u_\theta(x,T) - f(x) \right]^2 dx$$

Finally, we obtain the solution by optimizing these two loss functions.

$$u^*(x,t) = \underset{u_\theta(x,t)}{\arg\min} \left[ \lambda_1 \mathcal{L}_1 + \lambda_2 \mathcal{L}_2 \right]$$

where $\lambda_1$ and $\lambda_2$ are the weights of the two loss functions. $u_\theta(x,t)$ is the neural network with tanh activation function. Finally, we can obtain the expectation $u^*(x_0,0) = \mathbb{E}[f(X_T)|X_0 = x_0]$. All specific details regarding the implementation of the algorithms are provided in the appendix. The error analysis of this equation can be found in many works related to PINN, for example, in (De Ryck & Mishra, 2022).

**Discussion of the choice of the Feynman-Kac model** : The key to our approach lies in a change in the way expectations are calculated, utilizing the complete distributional information in $P$ contained in the approximation to obtain $(X_0^*, \mu^*, \sigma^*)$. However, in the approximation of $(X_0^*, \mu^*, \sigma^*)$ in many diffusion bridge models, it's often necessary to simulate part of the Brownian motion of the trajectory to estimate the loss function. This results in only some positions $(x,t)$ corresponding to $(\mu,\sigma)$ being accurate, while others rely on the network's generalization ability. Thus, in this case, the appearance of $x$ in our position $(x,t)$ occurs randomly, necessitating a meshless PDE solver. However, for certain $(\mu,\sigma)$ with exact analytical forms and diffusion bridges exhibiting better generalization, a non-meshless PDE solver may suffice. The second critical issue is the alteration in the way expectations are computed, introducing the dimension $d$ with respect to the MCMC expectation estimator. To overcome the curse of dimensionality, we require a PDE solver capable of handling this problem. For low-dimensional, non-meshless scenarios, finite element methods (Milstein et al., 2004) are viable. However, in more general cases, we require meshless PDE solvers that can address the curse of dimensionality. We have chosen a classical PDE solver called PINN, but other PDE solvers meeting these conditions are also feasible.

### 4.3 FEYNMAN-KAC OPERATOR EXPECTATION ESTIMATOR

FKEE is made up of two parts: Diffusion bridge model and Feynman-Kac model. The Diffusion bridge provides the coefficients and initial values of the SDE for the Feynman-Kac model. For a target distribution, we can first use the first model to obtain the corresponding coefficients and save them. The second model allows us to use these coefficients to directly approximate $\mathbb{E}_{X\sim P}[f(X)]$. Since the Feynman-Kac model is trained by using PINN, for high dimensional distributions, we can use GPU arithmetic acceleration or parallelism to get the corresponding results. This is different from the use of parallel Markov chains in MCMC algorithm to reduce the variance of the estimate. Our approach has a deterministic solution in the form of a PDE, so that the error is deterministic depending on the optimization algorithm and the training data.

## 5 DISCUSSION

In this section, we will discuss the scope of application of this method.

**A more detailed discussion of the $f$ and $P$ conditions**

**A discussion of $P$:** In conventional MCMC algorithms, such as Langevin diffusion, the characteristics of the potential energy function $V$ are extensively discussed. Specifically, the requirements for Lipschitz continuous gradients, strong convexity, and other details are detailed in (Cheng & Bartlett, 2018; Cheng et al., 2018). However, in our approach, the convergence and speed of the algorithm for solving the expectation do not hinge on the specific properties of $V$. Instead, we necessitate the corresponding SDE to have strong solutions and the Feynman-Kac equation to be well defined.

Regarding the applicability of our method, we can explore an alternative perspective. For the Itô type SDE as follows:

$$dX_t = \mu(X_t, t)dt + \sigma(X_t, t)dW_t, X_0 = x_0,$$

which corresponds to an Fokker–Planck–Kolmogorov (FPK) equation (Risken & Risken, 1996; Frank, 2005) as follows:

$$\frac{\partial p(x,t)}{\partial t} = -\sum_i \frac{\partial}{\partial x_i}\left[\mu_i(x,t)p(x,t)\right] + \frac{1}{2}\sum_{i,j}\frac{\partial^2}{\partial x_i \partial x_j}\left\{\left[\sigma(x,t)\sigma^\top(x,t)\right]_{ij}p(x,t)\right\},$$

where $p(x,t)$ is the probability density function for $X_t$. If we consider the stationary distribution, we set $\frac{\partial p(x,t)}{\partial t} = 0$. But there is more than one pair $(\mu, \sigma)$ for which this stationary FPK equation holds. Langevin diffusion is just a special case by setting $\sigma = I_d$ For other cases, which can be handled by our method, we add an example of a broader computation of the expectation of a stationary distribution in the absence of the corresponding convergence result for MCMC algorithm (Li, 2023).

$$dX_t = \frac{1}{2}h^2\frac{1 - 2X_t}{X_t^{\frac{1}{2}}(1 - X_t)^{\frac{1}{2}}}dt + 2hX_t^{\frac{1}{4}}(1 - X_t)^{\frac{1}{4}}dW_t,$$

stationary distribution is

$$p(x) = \frac{1}{Z}\frac{1}{\sqrt{2}x^{\frac{1}{4}}(1 - x)^{\frac{1}{4}}}.$$

$Z$ represents a normalization constant. Furthermore, our method can be applied to more general pairs $(\mu, \sigma)$ that satisfy the FPK equation, even in cases involving finite tiime and non-stationary conditions.

**A discussion of $f$:** MCMC typically employs two classical expectation estimators. One is based on the law of large numbers, often requiring Lipschitz continuity of $f$, and the estimator's variance is related to the Lipschitz coefficient of $f$. The other is based on the ergodic theorem, also imposing requirements on the Lipschitz coefficient, as well as the density function of $P$. Our method, on the other hand, only demands that the boundary conditions of the PDE satisfy a specific smoothness, namely $f \in C^2$. This significantly broadens the scope of this approach.

Differences from the classical MCMC expectation estimator due to the effect of $f$. In the classical MCMC expectation estimator we have the following two forms of computation:

$$\mathbb{E}\left[f(X)\right] = \frac{1}{N}\sum_{i=1}^{N}f(X_T^i),$$

where $X_T^i$ is the value at moment $T$ of the $i$th Markov chain. The different Markov chains are independent of each other. This one is also the classical calculation of the Monte Carlo integral. The error in this case is based on the law of large numbers and the central limit theorem. That is, for simple distributions such as the uniform distribution in $[0, 1]$, the bias occurs when $f(x) = x^n$ where $n$ is large. Another estimator applicable only when $P$ is in a stationary distribution.

$$\mathbb{E}\left[f(X)\right] = \frac{1}{N - M}\sum_{t=M}^{N}f(X_t).$$

In this scenario, averaging is performed over the time of a Markov chain. Here, $M$ represents the number of samples discarded during the burn-in period, where samples exhibit correlation. The error in this case is influenced by the ergodic theorem. However, determining an optimal value for $M$ can be challenging in complex problems. Introducing relevant samples can alleviate the impact on the efficiency of the MCMC expectation estimator.

Thus in difficult problems, often the properties of $f$ will tend to lead to larger biases. Whereas our approach greatly extends the efficiency of the MCMC method, on the one hand we use points inside the combustion period added to the loss of the PDE, and secondly we improve the assumptions on $f$, providing a new way to reduce the bias.

## 6    EXPERIMENTS

The first example we provide is about the computation of the partition function for random graph models. We simplify the experimental setup in the paper (Haddadan et al., 2021) by considering only the estimation of the mathematical expectation, and the estimation of the corresponding partition function. Backgrounds can be found in Appendix A.2.

First method: direct approximation of the overall part of the expectation. That is, we consider the approximate stochastic process $H_\beta(X)$, which is a one-dimensional problem. We generate the chain using the same method as in (Haddadan et al., 2021) and compute the value $H_\beta(X_t)$ under each moment. The diffusion bridge model and the Feynman-Kac model are then used to estimate the expectation separately. In the diffusion bridge model, we generated the same number of Brownian motions at the same number of moments and then calculated the loss at each moment to train. The Feynman-Kac model uses the already established diffusion bridge model to get an estimate of the expectation by solving the PDE.

The second approach better exemplifies the substantial improvement in harnessing Markov chains facilitated by our method. It highlights the remarkable flexibility embedded in our approach. Specifically, we directly approximate the distribution on a random graph, conceptualizing this graph as an $n^2$ random variable $(X_1, X_2, ..., X_{n^2})$, with each variable assuming two discrete values. A Markov chain is executed to obtain a sizable sample of random variables, and we subsequently approximate this $n^2$ dimensional distribution using a diffusion bridge model. However, since we are using a continuous model via SDE to obtain $Y_T$, which cannot accurately approximate a discrete random variable with values of $\{0, 1\}$, we employ the sigmoid function in the output $Y_T$. The loss for $X_T$ is then computed. Finally, when using the diffusion model, we apply post-processing to obtain the output value, i.e., $torch.round()$. In the case of the Feynman-Kac model, we set the boundary conditions to $u(x, T) = p(H(round(sigmoid(x))))$, where $p$ is $exp(-\beta/2 * (x))$. In other words, we set the composite function $p(H(round(sigmoid))$ to $f$ in the boundary.

In this experiment, our method demonstrates remarkable efficiency when handling distributions on random graphs, especially in the face of complexities introduced by the target expectation index and the function $H(x)$. For instance, in the high-dimensional scenario with $n = 15$, involving graphical models with partition functions summing over $2^{225}$ discrete points, our method proves effective. This enhancement becomes particularly evident when comparing $wi$ sample points, $vi$ sample points, and the MCMC estimator from (Haddadan et al., 2021). For further validation, please refer to `https://github.com/zysophia/Doubly_Adaptive_MCMC/blob/main/data/isingcompare_complexity.csv`. We achieve comparable accuracy using only 2000 points in the Markov chain, with reduced computation time compared to current state-of-the-art MCMC expectation estimators. The results of this experiment are presented in Appendix A.2.

This experiment focuses on the effect of $f$ on the expectation of the distribution of the target and the efficiency of the algorithm. Efficiency is defined here as the fact that the algorithm **uses fewer points on the Markov chain to achieve higher accuracy in approximating expectations**. Some baseline experiments on the properties of $P$ and low variance are presented in Appendix A.4.

## 7    CONCLUSION

In conclusion, we have introduced a novel approach for the estimation of Mathematical expectations and demonstrated impressive results with consistently low variance, even when dealing with limited sample sizes. Our method involves establishing a bridge between the realm of deep learning PDE solvers and the sampling domain, thereby enhancing the effectiveness of the MCMC expectation estimator while reducing reliance on conventional assumptions. We have introduced a versatile diffusion bridge model capable of utilizing either a partial sample or the density function of the target distribution to align with a diffusion bridge. Our work showcases its wide-ranging applicability across diverse domains, ensuring high efficiency through a single training process, and pushing the boundaries of the MCMC algorithm to achieve more effective utilization. Our contributions pave the way for more robust and versatile approaches to expectation estimation, and offer promise in various fields that rely on accurate and efficient probabilistic modeling.

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

# A APPENDIX

## A.1 DEFINITIONS AND RELATED THEORY

**Wasserstein distance:** The most commonly used measure of distance between probability distributions is the Wasserstein distance. It calculates the minimum cost of transporting mass from one distribution to another, based on the distance between the points being transported and the amount of mass being moved. The Wasserstein distance is especially beneficial for comparing distributions with different shapes since it considers the structure of distributions instead of only their statistical moments. This distance metric is widely applied in fields like image processing, computer vision, and machine learning. The definition is

$$W_p(\mu, \nu) := \left( \inf_{\pi \in \Pi(\mu, \nu)} \int_{\mathbb{R}^d \times \mathbb{R}^d} |x - y|^p \pi(dx, dy) \right)^{\frac{1}{p}} = \inf \left\{ [\mathbb{E}|X - Y|^p]^{\frac{1}{p}}, \mathbb{P}_x = \mu, \mathbb{P}_Y = \nu \right\}$$

$\Pi(\mu, \nu)$ denotes the class of measures on $\mathbb{R}^d \times \mathbb{R}^d$ with marginal distributions $\mu$ and $\nu$.

**Euler-Maruyama method:** (Platen, 1999) is a frequently used approach for solving SDE through an iterative format. This method has been shown to converge to a strong order of $\mathcal{O}(h^{\frac{1}{2}})$, where the error is dependent on the Lipschitz coefficients of the drift and diffusion coefficients. When generating paths using this method, it is recommended to use smaller step sizes, to minimize the errors associated with the method.

$$X_{t+h} = X_t + \mu(t, X_t)h + \sigma(t, X_t)(W_{t+h} - W_t), X_0 = x_0$$

Numerical solvers for stochastic differential equations of any accuracy are allowed when constructing sample paths for diffusion.

**Physics-informed neural networks:** PINN Raissi et al. (2019) is a deep learning method for solving partial differential equations. Main idea is to use neural networks for fitting solutions to PDE problems, PINN incorporates the residuals of the PDE (the difference between the left-hand side and the right-hand side of the PDE equation) into the loss function. and then updates the weights and parameters of the neural network through a backpropagation algorithm. Specifically, We consider follow PDE:

$$F(u_t, u_x, u_{xx}) = g(u, x, t)$$

and The boundary condition is

$$G(u_t, u_x, u_{xx}) = 0$$

We choose a neural network $u^\theta(x, t)$ to approximate the solution $u(x, t)$. By automatic differentiation, we can easily obtain the term $u_t^\theta, u_x^\theta$ and $u_{xx}^\theta$. We then need to sample the region of the target and calculate the value of the empirical loss function for these points. Finally the solution $u_t^\theta$ is obtained by optimising the combination of the two loss functions.

$$Loss\ PDE = F(u_t^\theta, u_x^\theta, u_{xx}^\theta) - g(u^\theta, x, t) \quad Loss\ boundary = G(u_t^\theta, u_x^\theta, u_{xx}^\theta)$$

$$Loss = \lambda_1 Loss\ boundary + \lambda_2 Loss\ PDE$$

$\lambda_1$ and $\lambda_2$ are the weights of the two loss functions.

**Feynman-Kac equation:**

If the following stochastic differential equation (SDE) has a strong solution, meaning that the drift and diffusion coefficients satisfy the conditions (Platen, 1999) below:

- Lipschitz condition

$$|\mu(x, t) - \mu(y, t)| \le K|x - y| \text{ and } |\sigma(x, t) - \sigma(y, t)| \le K|x - y|$$

  for all $t \in [0, T]$ and $x, y \in \mathbb{R}$

- Linear growth bound There exists a constant $C$ such that

$$|\mu(x, t)|^2 \le C^2(1 + |x|^2) \quad \text{and} \quad |\sigma(x, t)|^2 \le C^2(1 + |x|^2)$$

  for all $t \in [0, T]$ and $x, y \in \mathbb{R}$.

- Measurability

$$\mu(x,t) \text{ and } \sigma(x,t) \text{ is jointly measurable.}$$

- Initial value

$$X_0 \text{ is } \mathcal{F}_0\text{-measurable with } \mathbb{E}(|X_0|^2) < \infty.$$

Then, the solution to the corresponding backward partial differential equation (PDE) can represent the expectation of the terminal distribution of the SDE.

$$\frac{\partial u(x,t)}{\partial t} + \sum_{i=1}^{d} \frac{\partial u(x,t)}{\partial x_i} \mu_i(x,t) + \frac{1}{2} \sum_{i=1}^{d} \sum_{j=1}^{d} \frac{\partial^2 u(x,t)}{\partial x_i \partial x_j} (\sigma(x,t)\sigma(x,t)^T)_{i,j} = 0$$

$$u(x,T) = f(x).$$

The soluton of PDE is $u(x_0, 0) = \mathbb{E}[f(X_T)|X_0 = x_0]$

**Proof:** According to Itô's formula

$$du(X_t, t) = \left[ \frac{\partial u(X_t, t)}{\partial t} + \sum_{i=1}^{d} \frac{\partial u(X_t, t)}{\partial x_i} \mu_i(X_t, t) + \frac{1}{2} \sum_{i=1}^{d} \sum_{j=1}^{d} \frac{\partial^2 u(X_t, t)}{\partial x_i \partial x_j} (\sigma(X_t, t)\sigma(X_t, t)^T)_{i,j} \right] dt$$

$$+ \left[ \sum_{r=1}^{d} \sum_{i=1}^{d} \frac{\partial u(X_t, t)}{\partial x_i} \sigma_{i,r}(X_t, t) \right] dW_t^r,$$

where $W_t^r$ is the rth component of $W_t$. The first part is based on the equality inside the PDE being set to zero. Integrate from $[0,T]$ on both sides.

$$u(X_T, T) - u(X_0, 0) = f(X_T) - u(X_0, 0) = \int_0^T \left[ \sum_{r=1}^{d} \sum_{i=1}^{d} \frac{\partial u(X_t, t)}{\partial x_i} \sigma_{i,r}(X_t, t) \right] dW_t^r,$$

Taking the conditional expectation on both sides while fixing $X_0$ and utilizing the properties of Itô integration as a martingale.

$$u(x_0, 0) = \mathbb{E}[f(X_T)|X_0 = x_0]$$

## A.2 BACKGROUNDS AND TABLES OF THE EXPERIMENT

**Ising model**: Assume a sample space $\Omega$, Hamiltonian function $H : \Omega \rightarrow \{0\} \cup [1, \infty)$, and inverse temperature parameter $\beta \in \mathbb{R}$, referred to as inverse temperature. The Gibbs distribution on $\Omega, H(\cdot)$, and $\beta$ is then characterized by probability law $\forall x \in \Omega : \pi_\beta(x) \doteq \frac{1}{Z(\beta)} \exp(-\beta H(x))$ Here $Z(\beta)$ is the normalizing constant or Gibbs partition function (GPF) of the distribution, with $Z(\beta) \doteq \sum_{x \in \Omega} \exp(-\beta H(x))$. Specifically, we considered Ising model on 2D lattices: It has $n \times n$ dimensions and a total of $n^2$ random variables, each of which takes the values +1,-1 with Hamiltonian function $H(x) = -\sum_{(i,j) \in E} \mathbb{I}(x(i) = x(j))$. For $\beta_0$ the results are easy to compute and for $[\beta_1, \beta_2]$ between we can use the PPE-method, we did not use the Tpa-Method (Haddadan et al., 2021), which is an algorithm on splitting the region $[\beta_1, \beta_2]$. Specifically we can compute the following $\mathbb{E}F = \mathbb{E}\exp(-\frac{\beta_2 - \beta_1}{2} H(X_{\beta_1}))$ and $\mathbb{E}G = \mathbb{E}\exp(\frac{\beta_2 - \beta_1}{2} H(X_{\beta_2})).X_{\beta_i}$ is a Gibbs distribution obeying parameter $\beta_i$. $Q = \frac{\mathbb{E}G}{\mathbb{E}F} = \frac{Z(\beta_1)}{Z(\beta_2)}$.We set $\beta_1 = -0.02$ and $\beta_2 = 0$. Then we can find $Z(\beta_1)$ based on the fact that $Z(\beta_2) = Z(0)$. So we need to estimate two mathematical expectations and we propose two ways to approximate this expectation. In this Experiment, for a definite temperature $\beta$, the distribution on the random graph is often easy to approximate, but the complexity of the exponent in the target expectation and also the function $H(x)$ can lead to the need for a large sample size to reduce the variance when MCMC deals with this problem. Our approach demonstrates superior efficiency in dealing with the distribution on a random graph, particularly when considering the complexities introduced by the target expectation exponent and the function $H(x)$. The implementation of our method, FKEE, stands out in handling larger-sized graphs ($n \geq 6$) where traditional MCMC and its variants, as found in (Haddadan et al., 2021), face challenges due to sample complexity.

Table 2 and Table 3 are one table. We have separated them for ease of presentation, and they have the same rows. In Table 2 and Table 3, $wi, vi, z$ represent the values of the corresponding $\mathbb{E}F, \mathbb{E}G, Q$ estimated using the corresponding estimators, respectively. $true\_wi, true\_vi, true\_z$ indicate the corresponding true values. The $error\_wi, error\_vi, error\_z$ represent the squared error using the corresponding estimators. The terms $w_i$ sample points and $v_i$ sample points refer to the number of sampled points utilized by the estimator. The terms $w_i$ time and $v_i$ time refer to the time taken by the estimator, measured in seconds. MCMC method we employed to generate samples follows the same approach as used in `https://github.com/zysophia/Doubly_Adaptive_MCMC`.

At the same time we compare with the method RelMeanEst in (Haddadan et al., 2021). MCMC-C is the method RelMeanEst, MCMC-R is the empirical mean taken using the samples obtained from resampling, and MCMC-T is the estimate of the expectation obtained using the established diffusion bridge. And the number of data points used indicates the number of points in the Markov chain used. To be fair, we lower the threshold in MCMC-C to reduce its algorithmic complexity. Because only a small number of sample points are used in MCMC-R and MCMC-T. sample points means the number of points sampled from the Markov chain. Note that when $n \geq 6$ is in the MCMC-C method due to the larger complexity we do not discuss it. We only compare MCMC-R and MCMC-T. Note: GPU types: the first of these uses Tesla P100 while the second uses Tesla V100 when $n \geq 6$. The two methods are shown in Table 2 and Table 3. Above the horizontal is the first method below the second method. We can find the performance of PINN. In the high-dimensional case ($d = n^2 = 225$).

Table 2: Comparison of different MCMC Expectation Estimator

| Method | $n$ | $wi$ | $vi$ | $z$ | $true\_wi$ | $true\_vi$ | $true\_q$ |
|---|---|---|---|---|---|---|---|
| MCMC-C | 2 | 0.9706396 | 1.0306606 | 1.0618365 | 0.9654024 | 1.0357122 | 1.072778 |
| MCMC-R | 2 | 0.9550395 | 1.0308957 | 1.0794273 | 0.9654024 | 1.0357122 | 1.072778 |
| MCMC-T | 2 | 0.9626546 | 1.0333116 | 1.073398 | 0.9654024 | 1.0357122 | 1.072778 |
| MCMC-C | 3 | 0.9340726 | 1.0744393 | 1.1502738 | 0.9226402 | 1.0834867 | 1.174333 |
| MCMC-R | 3 | 0.9269992 | 1.0774463 | 1.1622948 | 0.9226402 | 1.0834867 | 1.174333 |
| MCMC-T | 3 | 0.9283546 | 1.0795156 | 1.1628268 | 0.9226402 | 1.0834867 | 1.174333 |
| MCMC-C | 4 | 0.8844253 | 1.1470378 | 1.2969301 | 0.8641533 | 1.1563625 | 1.338233 |
| MCMC-R | 4 | 0.8686283 | 1.159192 | 1.3345087 | 0.8641533 | 1.1563625 | 1.338233 |
| MCMC-T | 4 | 0.8692993 | 1.1552249 | 1.328915 | 0.8641533 | 1.1563625 | 1.338233 |
| MCMC-R | 2 | 0.9950697 | 1.00498 | 1.0099593 | 0.9654024 | 1.0357122 | 1.072778 |
| MCMC-T | 2 | 0.9735975 | 1.0444663 | 1.0727907 | 0.9654024 | 1.0357122 | 1.072778 |
| MCMC-R | 3 | 0.9949918 | 1.0049603 | 1.0100187 | 0.9226402 | 1.0834867 | 1.174333 |
| MCMC-T | 3 | 0.9188372 | 1.0843412 | 1.1801233 | 0.9226402 | 1.0834867 | 1.174333 |
| MCMC-R | 4 | 0.9951742 | 1.0050679 | 1.0099418 | 0.8599499 | 1.1563472 | 1.344668 |
| MCMC-T | 4 | 0.8664092 | 1.1570783 | 1.3354871 | 0.8599499 | 1.1563472 | 1.344668 |
| MCMC-R | 6 | 0.9949221 | 1.0049597 | 1.0100888 | 0.7163408 | 1.3985122 | 1.9523 |
| MCMC-T | 6 | 0.6953082 | 1.3970394 | 2.0092378 | 0.7163408 | 1.3985122 | 1.9523 |
| MCMC-R | 8 | 0.9950855 | 1.0050602 | 1.010024 | 0.5468445 | 1.8348543 | 3.3553494 |
| MCMC-T | 8 | 0.5683886 | 1.9384431 | 3.4104189 | 0.5468445 | 1.8348543 | 3.3553494 |
| MCMC-R | 10 | 0.9949943 | 1.0050541 | 1.0101104 | 0.3853279 | 2.60382 | 6.7574135 |
| MCMC-T | 10 | 0.3684352 | 2.833073 | 7.6894751 | 0.3853279 | 2.60382 | 6.7574135 |
| MCMC-R | 15 | 0.9949888 | 1.0050285 | 1.0100903 | 0.1135434 | 8.894777 | 78.3381355 |
| MCMC-T | 15 | 0.1181741 | 10.4130456 | 88.1161667 | 0.1135434 | 8.894777 | 78.3381355 |

### A.3 APPROXIMATION OF TWO TARGET DISTRIBUTIONS AND ALGORITHMS

The algorithm of the Feynman-Kac model is similar to that of PINN. It is mainly a matter of using the diffusion coefficients obtained earlier and solving the corresponding PDE for the data points. Not all points on the paths need to be included in the training in this algorithm. This is the same as the training of PINN, where we only need to sample a fraction of the points to get the solution. This Algorithms is known as the diffusion bridge model, and it is applicable in scenarios where the target distribution is unknown, but there exist high-quality sampling points, or when it's necessary to match a given Markov chain.

Table 3: Comparison of different MCMC Expectation Estimator

| $error\_wi$ | $error\_vi$ | $error\_q$ | $wi$ sample points | $vi$ sample points | $wi$ time (s) | $vi$ time (s) |
|---|---|---|---|---|---|---|
| 2.74E-05 | 2.55E-05 | 0.000119716 | 3157 | 3157 | 0.29448 | 0.28554 |
| 0.00010739 | 2.32E-05 | 4.42E-05 | 100 | 100 | 14.421 | 9.702 |
| 7.55E-06 | 5.76E-06 | 3.84E-07 | 100 | 100 | 11.671 | 11.789 |
| 0.0001307 | 8.19E-05 | 0.000578845 | 30700 | 30700 | 3.40208 | 3.43238 |
| 1.90E-05 | 3.65E-05 | 0.000144918 | 2000 | 2000 | 20.163 | 19.544 |
| 3.27E-05 | 1.58E-05 | 0.000132393 | 2000 | 2000 | 236.109 | 233.225 |
| 0.000410954 | 8.70E-05 | 0.00170593 | 11383 | 11383 | 0.922 | 0.93 |
| 2.00E-05 | 8.01E-06 | 1.39E-05 | 2000 | 2000 | 26.917 | 26.696 |
| 2.65E-05 | 1.29E-06 | 8.68E-05 | 2000 | 2000 | 284.784 | 285.609 |
| | | | | | | |
| 0.000880149 | 0.000944468 | 0.003946189 | 500 | 500 | 7.562 | 5.728 |
| 6.72E-05 | 7.66E-05 | 1.61E-10 | 2000 | 2000 | 15.809 | 15.456 |
| 0.005234754 | 0.006166395 | 0.026999189 | 500 | 500 | 7.748 | 7.547 |
| 1.45E-05 | 7.30E-07 | 3.35E-05 | 2000 | 2000 | 32.482 | 32.562 |
| 0.018285611 | 0.022885427 | 0.112041629 | 500 | 500 | 10.762 | 10.573 |
| 4.17E-05 | 5.35E-07 | 8.43E-05 | 2000 | 2000 | 59.597 | 58.064 |
| 0.077607541 | 0.15488357 | 0.887761945 | 500 | 500 | 25.194 | 23.99 |
| 0.00044237 | 2.17E-06 | 0.003241913 | 2000 | 2000 | 302.916 | 303.36 |
| 0.200919994 | 0.688558248 | 5.500551232 | 500 | 500 | 64.449 | 62.8 |
| 0.000464148 | 0.010730639 | 0.00303265 | 2000 | 2000 | 516.126 | 509.603 |
| 0.371693119 | 2.556052403 | 33.03149292 | 500 | 500 | 91.826 | 90.796 |
| 0.000285363 | 0.052556938 | 0.868738826 | 2000 | 2000 | 889.684 | 906.444 |
| 0.776945993 | 62.24813139 | 5979.626574 | 500 | 500 | 165.086 | 166.38 |
| 2.14E-05 | 2.305139542 | 95.60989415 | 2000 | 2000 | 1968.976 | 1997.663 |

---

**Algorithm 1** Diffusion bridge model (DBM)

---

**Input:** epochs:$M$,Total point in time:$D$,Learning Rate:$r$,Initial value:$X_0$,Brownian motion :$W_t$
    Time Series: $t_0, t_1, \ldots, t_D = T$. Neural network: $\mu_{\theta_1}(x,t), \sigma_{\theta_2}(x,t), X_{0,\theta_3}$ and $\theta$ is the parameter of a neural network. Euler-Maruyama method of step $h$. Number of pathss simulated $N$. $\varepsilon$ is the required error threshold. The given data point is $Y_T$.
**Output:** $X_i, \mu(t, X_i), \sigma(t, X_i), i \in [t_0, t_1, \ldots, t_D]$
 1: Calculate $X_t$

$$X_{t+h} = X_t + \mu_{\theta_1}(t, X_t)h + \sigma_{\theta_2}(t, X_t)(W_{t+h} - W_t) \quad X_0 = X_{0,\theta_3}$$

 2: **for** $k$ in $1 : M$ **do**
 3:     Calculate loss
$$\mathcal{L} = W_p(\hat{\mathbb{P}}(X_T), \hat{\mathbb{P}}(Y_T)))$$

 4:     **if** Match the whole Markov chain **then**
 5:         Calculating the loss of this path assumes a Markov chain with $M$ steps

$$\mathcal{L} = \sum_{i=1}^{M} W_p(\hat{\mathbb{P}}(X_i), \hat{\mathbb{P}}(Y_i)))$$

 6:     **end if**
 7:     **for** $n$ in $1 : 3$ **do**
 8:         Update parameters $\theta_n^k \leftarrow \theta_n^{k-1} - \nabla_\theta \mathcal{L} * r$
 9:     **end for**
10:     **if** $\mathcal{L} < \varepsilon$ **then** Break
11:     **end if**
12: **end for**

---

**Algorithm 2** Feynman-Kac model (FCM)

---

**Input:** epochs:$M$ ,Total point in time:$D$ ,Learning Rate: $r$ ,Time Series: $t_0, t_1, \ldots, t_D = T$. Points of observation :$X_t$,Drift coefficient: $\mu(t, X_t)$,Diffusion coefficient:$\sigma(t, X_t)$ where $t \in [t_0, t_1, \ldots, t_D]$Neural network: $u_\theta(x, t)$ $\theta$ is the parameter of a neural network. The function $f$ that needs to be estimated. Number of paths simulated $N$. required error threshold $\varepsilon$.

**Output:** $\mathbb{E}(f(X_T)|X_0 = x_{t_0}) = u_\theta(x_{t_0}, t_0)$

    **for** $k$ in $1 : M$ **do**

        **if** $\sigma(t, X_t) = I_{d \times d}$ **then**

            **for** $s$ in $1 : D - 1$ **do**

$$\mathcal{L}_1^s = \frac{1}{N} \sum_{k=1}^{N} \left\{ \frac{\partial u_\theta(x,t)}{\partial t} + \sum_{i=1}^{d} \frac{\partial u_\theta(x,t)}{\partial x_i} \mu_i(x_t, t) + \frac{1}{2} \sum_{i=1}^{d} \frac{\partial^2 u_\theta(x,t)}{\partial x_i^2} \Bigg|_{(x,t)=(x_s^k, t_s)} \right\}^2$$

            **end for**

        **end if**

        **if** $\sigma(t, X_t) \neq I_{d \times d}$ **then**

            **for** $s$ in $1 : D - 1$ **do**

$$\mathcal{L}_1^s = \frac{1}{N} \sum_{k=1}^{N} \left\{ \frac{\partial u_\theta(x,t)}{\partial t} + \sum_{i=1}^{d} \frac{\partial u_\theta(x,t)}{\partial x_i} \mu_i(x, t) + \frac{1}{2} \sum_{i=1}^{d} \sum_{j=1}^{d} \frac{\partial^2 u_\theta(x,t)}{\partial x_i \partial x_j} (\sigma(x,t)\sigma(x,t)^T)_{i,j} \Bigg|_{(x,t)=(x_s^k, t_s)} \right\}^2$$

            **end for**

        **end if**

        Calculate PDE loss

$$\mathcal{L}_1 = \sum_{s=1}^{D-1} \mathcal{L}_1^s$$

        Calculate boundary loss

$$\mathcal{L}_2 = \frac{1}{N} \sum_{k=1}^{N} \left\{ u_\theta(x_{t_D}^k, t_D) - f(x_{t_D}^k) \right\}^2$$

        Update parameters $\theta^k \leftarrow \theta^{k-1} - \nabla_\theta(\mathcal{L}_1 + \mathcal{L}_2) * r$

        **if** $(\mathcal{L}_1 + \mathcal{L}_2) < \varepsilon$ **then** Break

        **end if**

    **end for**

---

Since there are too many variants for the MCMC sampler, and our aim in this paper is to estimate the expectation rather than focusing on the selection aspect of the sampler, we consider one of the simplest LMCMC (Langevin diffusion model). It is worth noting, however, that we are using the unadjusted LMCMC here.

---

**Algorithm 3** Unadjusted Langevin diffusion model (LDM)

---

**Input:** Total point in time:$D$,Initial value :$X_0 = x_0$,Brownian motion :$W_t$ Time Series: $t_0, t_1, \ldots, t_D = T$. Euler-Maruyama method of step $h$. Number of paths simulated $N$. The distribution of the given data is a $p(x) = \frac{1}{Z} \exp(-V(x))$.
**Output:** $X_i, \mu(X_i), i \in [t_0, t_1, \ldots, t_D]$
 1: Calculate $X_t, \mu(x_t)$

$$\mu(x) = \frac{1}{2}\nabla_x \log p(x) \quad X_{t+h} = X_t + \mu(X_t)h + W_{t+h} - W_t \quad X_0 = x_0$$

---

### A.4 Other baseline Experiments

We consider the MCMC algorithm and our method to sample the density function of the target and obtain the corresponding expectation. In the MCMC algorithm configuration, we use the Langevin MCMC to get independent samples. Here we use only the value of $X_T$ at the terminal moment to estimate the expectation. We use the same paths in LDM+FCM, but with a different way of computing expectations.

As an illustration, we consider a one-dimensional SDE, where we define the target distribution as $p(x) = C \exp(\frac{-(x-1)^2}{2})$, corresponding to the drift coefficient of the LDM being $\mu(x) = \frac{1-x}{2}$. We evaluate the expectation of $\mathbb{E}(X_{10})$. To decrease the error of the Euler-Maruyama method, we use a small step size of $h = 0.01$ and iterate 1000 steps to obtain the final path. We repeat the experiment $M = 30$ times. During the training process, we extract points from each path every 100 points and add them to the training process, instead of using all the points on the path.

We examine an extreme case, employing a very limited number of paths ($N = 5$) to estimate the true expectation $\mathbb{E}(X_{10})$. In Figure 1, we present the empirical distributions obtained through two different methods. The results obtained by LDM+FCM outperform Langevin MCMC, validating that paths can offer more informative outcomes. By incorporating gradient information from path points and integrating it into PINN for training, our method demonstrates lower variance under the same experimental configuration, significantly enhancing the efficiency of the MCMC algorithm with appropriate optimization. Although we utilize Unadjusted Langevin MCMC, our method provides unbiased estimates. **This is attributed to the fact that the bias in Unadjusted Langevin MCMC stems from the numerical SDE solver, while our method does not necessitate high accuracy in $X_t$; we are more concerned with the precision of the corresponding $(\mu, \sigma)$ on $X_t$.** Unlike direct sampling using the SDE method, which requires a highly precise SDE solver (Mou et al., 2021), such precision is unnecessary in our method. We only require accurate estimations at each point on the path for the coefficients of the drift and diffusion terms.

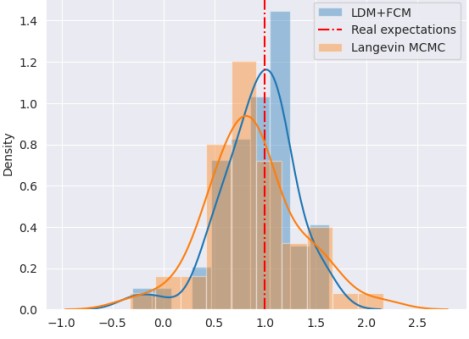

Figure 1: The empirical distribution of $\mathbb{E}_{estimated}(X_{10})$

For other cases, which can be handled by our method, we add an example of a broader computation of the expectation of a stable distribution in the absence of the corresponding convergence result for MCMC. for example:

$$dX_t = \frac{1}{2}h^2 \frac{1 - 2X_t}{X_t^{\frac{1}{2}}(1 - X_t)^{\frac{1}{2}}} dt + 2hX_t^{\frac{1}{4}}(1 - X_t)^{\frac{1}{4}} dW_t$$

where $X_0 = 0.5, \quad \mathbb{E}X_1 = 0.5$

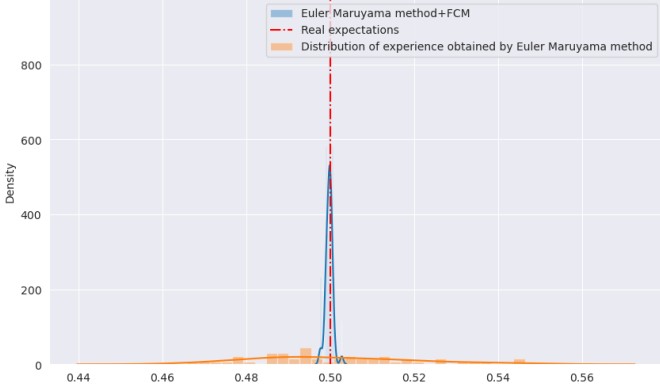

Figure 2: The empirical distribution of $\mathbb{E}_{estimated}(X_1)$

In this example, we use this method to estimate mathematical expectations in high dimensions. We consider a normal distribution with independent and identical marginal distributions as follows: $p(x) = C \exp(-0.5(x - 0.2)^2)$ for each dimension. When $g(x_1, \ldots, x_d) = x_1 + \cdots + x_d$, We use a smaller number of paths ($N = 50, 100$). We use the Euler-Maruyama method with a step size of 0. 1 for 100 iterations and calculate the internal loss function of the PDE every 10 points. In the case of $d = 5, 10$, we employ a 2-layer neural network with 108 units per layer and a tanh activation function. In the case of $d = 20$, we use a 2-layer neural network with 526 units per layer, set $N = 100$, and compute the internal loss function of the PDE every 20 points. We also repeat the experiment $M = 30$ times by using different random number seeds and measure the average time required to estimate the mathematical expectation each time with GPU (Tesla P100). In the training process we train 400 epochs by using the Adam optimizer with a learning rate of 0.001.

We compute $\mathbb{E}(g(X_1, X_2, \ldots, X_d))$ where $X_i \sim N(0.2, 1)$ and estimate its error. The error we use is

$$\textbf{Absolute value error} = \frac{1}{M} \sum_{i=1}^{M} |\mathbb{E}_{estimated}^i(g(X_1, X_2, \ldots, X_d)) - \mathbb{E}(g(X_1, X_2, \ldots, X_d)))|$$

$$\textbf{Square Error} = \frac{1}{M} \sum_{i=1}^{M} |\mathbb{E}_{estimated}^i(g(X_1, X_2, \ldots, X_d)) - \mathbb{E}_{mean}(g(X_1, X_2, \ldots, X_d)))|^2$$

where

$$\mathbb{E}_{mean}(g(X_1, X_2, \ldots, X_d))) = \frac{1}{M} \sum_{i=1}^{M} \mathbb{E}_{estimated}^i(g(X_1, X_2, \ldots, X_d))$$

The method is LDM+FCM and we compare the results of this method with those obtained by the Langevin MCMC (LMCMC in short).

This method is also applicable for estimating integrals in high dimensions, particularly in tandem with the Monte Carlo method, when the target distribution is easily samplable. We conducted a comparison in estimating the distribution of the bridge using a bridge constructed from partially sampled high-quality samples. This approach enables continuous sampling of the target distribution by utilizing a well-established bridge. To illustrate, we simulated a diffusion bridge model (DBM) to approximate the distribution of a target variable $Y = (X_1, X_2, X_3)$, where $X_1 \sim N(1, 2) + Beta(4, 2)$,

Table 4: Comparison of different methods

| Method | Dimension($d$) | paths($N$) | Absolute value error | Square Error | GPU time |
|--------|---------------|-----------|---------------------|--------------|----------|
| LMCMC | 5 | 50 | 2. 927620e-01 | 1.253495e-01 | $\times$ |
| LDM+FCM | 5 | 50 | 1. 031084e-01 | 1.600998e-02 | 29. 62s |
| LMCMC | 10 | 50 | 4. 696985e-01 | 3.077161e-01 | $\times$ |
| LDM+FCM | 10 | 50 | 3. 330310e-01 | 1.382318e-01 | 46. 79s |
| LMCMC | 20 | 100 | 3. 630368e-01 | 1.863313e-01 | $\times$ |
| LDM+FCM | 20 | 100 | 2. 959023e-01 | 1.042063e-01 | 49. 74s |

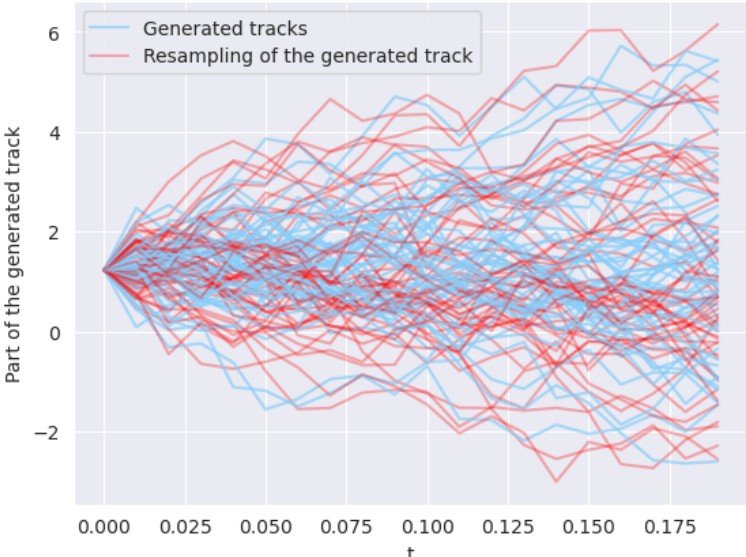

Figure 3: Generated tracks

$X_2 \sim N(-1, 2) + Gamma(1, 2)$, and $X_3 \sim N(3, 2) + geometric(0.5)$. We sampled 500 points from the target distribution and employed DBM matching to obtain an SDE. Subsequently, we compared the distribution of the generated tracks to the target distribution. Continuing the target distribution sampling using the constructed bridge, we sampled an additional 500 points and compared the differences between the resampled samples and the original target distribution. The specific parameters include $T = 0.2$, step size $h = 0.025$. The Diffusion Flow Model (DFM) is trained for 300 epochs with a learning rate of 0.001, using the Adam optimizer and Wasserstein distance as the loss. This method facilitates the construction of a pair of target distributions amenable to sampling. The expectation $\mathbb{E}(f(X))$ of the target distribution can be obtained by utilizing FCM.

## A.5 POTENTIAL APPLICATIONS AND FUTURE WORK

**The independence of samples:** The independence of samples plays a crucial role in machine learning, and its violation can significantly impact the performance and validity of machine learning models. Many machine learning models rely on the assumption of independent and identically distributed (i.i.d) samples. Non-independent samples can introduce dependencies that the model may mistakenly learn as patterns. Nonlinear mathematical expectations play a critical role in such non-iid scenarios (Peng, 2010). However, methods like using Max-Mean Monte Carlo for calculating nonlinear mathematical expectations are often challenging. This is because we need to partition the dataset into parts where the samples are independent and then calculate the linear mathematical expectation for each part. Finally, we take the largest to get the nonlinear mathematical expectation. Our approach provides a completely new way to consider the use of Stochastic Differential Equations (SDEs) with G-Brownian motions. The diffusion bridge model is constructed using the same method and then solved directly using the Feynman-Kac model in the case of nonlinear mathematical expectations. This avoids problems such as data grouping.

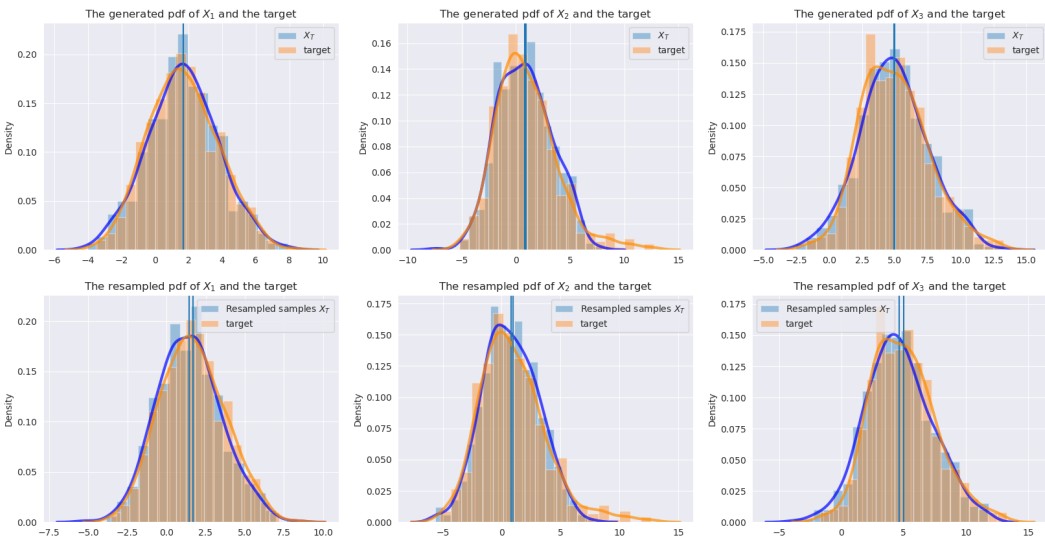

Figure 4: Comparison of the probability density functions of the generated and resampled paths and target distributions for each dimension. Two of the blue lines are the mean of the experience of the target sample and the mean of the experience of the re-generation sample, respectively.

**Representation learning and Distributional regression learning:** In the theory of statistical learning, we assume $X \sim P_X$ and $Y \sim P_Y$. A basic loss function is $l = \mathbb{E}[h_\theta(X) - Y)^2)]$ and $l = \mathbb{E}[\textbf{Corss Entropy}[h_\theta(X), Y)]$ where $h_\theta$ is model, and we often need to sample a portion of the sample $\{x_i, y_i\}_{i=1}^N$, and then optimise the empirical loss function $l = \frac{1}{N}\sum_{i=1}^N (h_\theta(x_i) - y_i)^2$ and $l = \frac{1}{N}\sum_{i=1}^N \textbf{Corss Entropy}(h_\theta(x_i), y_i)$. But in the case where the sample size does not fully cover the distribution of the corresponding totality, because the loss function is obtained by sampling a portion of the dataset, the loss function that we obtain tends to be biased, or has a large variance. When we have a high quality diffusion bridge that can accurately approximate the distribution of the target $(P_X, P_Y)$, which most of the current diffusion bridge models can do. We can achieve this by configuring the boundary conditions in the Feynman-Kac model to be $f(x, y) = (h_\theta(x) - y)^2$ and **Cross Entropy**$(h_\theta(x), y)$. We then replace the empirical loss function with the PDE loss and the PDE loss at the boundary. This approach may enable us to enhance the learning of the **Representation of a Distribution**. This is because the diffusion bridge model captures information about the entire distribution rather than just the local distribution of specific points. When estimating expectations, we incorporate the PDE loss function, which contains gradient information regarding the diffusion bridge coefficients. The coefficients of the diffusion bridge tend to exhibit correlations with the target distribution. In this case, the number of points required for the diffusion bridge coefficients is often significantly smaller than the number of points $N$ directly sampled from the data. Finally, we can use the trained diffusion bridge model to perform some basic statistical learning tasks.

**Variational Inference:** Due to the extensive application of mathematical expectations in machine learning and probabilistic statistics, we are unable to comprehensively demonstrate all relevant methods in this paper. We will consider applying these methods to important domains, such as estimating the evidence lower bound (**ELBO**) in Black-Box Variational (Ranganath et al., 2014) Inference. We often need to use **reparameterization** techniques to estimate the **ELBO** $= \mathbb{E}_{q(z|\phi)} \log p(x, z) - \log q(z|\phi)$ with small bias, but we can consider using a diffusion bridge to approximate the target distribution $q(z|\phi)$, and select $f(z) = \log p(x, z) - \log q(z|\phi)$, where $\phi$ can be designed as a trainable parameter. In this way, we can modify our optimization objective from **ELBO** to $-u(x_0, t_0) + \texttt{PDE loss} + \texttt{boundary loss}$, which can achieve lower variance and GPU acceleration.

