# OpenReview forum: "Feynman-Kac Operator Expectation Estimator: An Innovative Method for Enhancing MCMC Efficiency and Reducing Variance"
_ICLR.cc/2024/Conference — Submitted to ICLR 2024_

### Official Review · Reviewer_Q5pZ · 2023-10-30

**Soundness:** 3 good
**Presentation:** 1 poor
**Contribution:** 2 fair
**Rating:** 3
**Confidence:** 4

**Summary:**

This paper introduced a Feynman-Kac model approach for estimating mathematical expectations.

**Strengths:**

The proposed approach can potentially lead to more accurate estimations for mathematical expectations.

**Weaknesses:**

1. The paper's presentation is poor, lacking the algorithm and numerical results in the main text. It makes the proposed method inaccessible to the readers.

2. Some notation is used in Algorithm 1 without proper definition, e.g., MSE(.,0).

3. The proposed method lacks theoretical guarantees.

**Questions:**

1. How is MSE(., 0) is defined?

2. Add more explanations or theoretical proofs for the statement that ``the solution to the Feynman-Kac equation at the initial time is E(f(X_T)|X_0=x_0).

---

> ### Author Response · Authors · 2023-11-11
> **Reply to the Reviewer Q5pZ**
>
> Q1 The paper's presentation is poor, lacking the algorithm and numerical results in the main text. It makes the proposed method inaccessible to the readers.
>
> A1 The detailed algorithm and numerical results are indeed provided in the appendix.
>
> Q2 Some notation is used in Algorithm 1 without proper definition, e.g., MSE(.,0).
>
> A2 We have addressed this issue in the revised version.
>
> Q3 Add more explanations or theoretical proofs for the statement that ``the solution to the Feynman-Kac equation at the initial time is $E(f(X_T)|X_0=x_0)$.
>
> A3 We have addressed this issue in the revised version. This is a fundamental conclusion, and a concrete proof can be found at [1]
> [1] Applied stochastic differential equations. S Särkkä, A Solin - 2019 - Cambridge University Press .page 118

---

### Official Review · Reviewer_crQM · 2023-10-30

**Soundness:** 1 poor
**Presentation:** 1 poor
**Contribution:** 2 fair
**Rating:** 3
**Confidence:** 3

**Summary:**

This paper proposes a connection of MCMC expectation estimators with the Feynman-Kac equation and proposes to solve a bridge-matching problem to estimate the MCMC expectation.

**Strengths:**

* The paper proposed a novel approach to connect partial differential equations and Markov chain Monte Carlo.

**Weaknesses:**

The paper does not contain any of the experimental results in the main submission. The main text should be self-contained and present at least part of the representative results. Remember that the reviewers are not obligated to look into the supplementary material.

The current state of writing makes it very hard to grasp the gist of the proposed methodology. Let me elaborate on a few examples:
* For Section 2, the related work mixes up MCMC methods based on simulating the Langevin diffusion, score function modeling based on Langevin diffusion, and bridge matching. These are very different ideas for solving different problems. Thus, the way this section is written confused me rather than putting the work in the context of previous works.
* In Section 3.1, a key aspect of the methodology seems to solve a bridge-matching problem with respect to the samples of some MCMC chains. However, I could not find the specifics of the authors’ approach anywhere in the main text. Only after looking at Algorithm 1 in the appendix I found that some loss functions are being minimized. The definition of these loss functions should have been in the main text.
* For Section 3.2, the authors do not provide enough explanation/motivation/proof as to why the reverse solution of the Feynman-Kac equation is the right conditional expectation. Again, the paper is expected to be self-contained to a certain degree.

Furthermore, I have some concerns about the technical claims of the paper:
* The paper claims that the proposed methodology "necessitates fewer assumption since it does not rely on the law of large numbers and the Markov ergodic theorem." However, it does rely on a strong assumption: that we are able to solve the PDE! That is, the paper does not contain any proof of the accuracy of the solution of the proposed differential equation. Even if, mathematically, the differential equation's solution is the desired conditional expectation, discretizing and numerically solving this is a completely different story. This is crucial in an MCMC setting since empirically assessing the quality of samples/expectation estimates is very challenging.
* One of the key motivations seems to be replacing the need for discarding burn-in samples. However, I do not quite understand how this is true when the paper tries to match the output of an MCMC estimator. This will contain non-asymptotic bias that can only be resolved through burn-in. Then wouldn't trying to match this end up inheriting the bias?
* The paper claims that the method "enhances efficiency through one-time training." However, I do not quite see how this improves anything. Couldn't I run MCMC longer instead of solving a differential equation? In fact, if we assume that solving differential equations indeed provides the right expectation, couldn't MCMC be regarded as an efficient way to solve this equation without involving any solvers? How much faster, precisely, is solving the equation rather than running MCMC?

Lastly, the idea of connecting MCMC expectation estimators with partial differential equations is not new [1-4], and a literature is already building around it. While the approaches of these works are different, and I'm not saying that this work lacks novelty in that regard, I believe these works actually do what the paper seems to have attempted to do: post-process the output of an MCMC algorithm to improve the performance of the expectation estimator. Furthermore, they come with asymptotic and non-asymptotic consistency guarantees and work incredibly well in practice! Thus, I believe the paper should have compared and discussed these methods.

### References
Disclaimer: I am not the author nor affiliated with any of the papers below.
1. Oates, Chris J., Mark Girolami, and Nicolas Chopin. "Control functionals for Monte Carlo integration." Journal of the Royal Statistical Society Series B: Statistical Methodology 79.3 (2017): 695-718.
2. Sun, Zhuo, Alessandro Barp, and François-Xavier Briol. "Vector-valued control variates." International Conference on Machine Learning. PMLR, 2023.
3. South, Leah F., et al. "Semi-exact control functionals from Sard’s method." Biometrika 109.2 (2022): 351-367.
4. South, Leah F., et al. "Regularized zero-variance control variates." Bayesian Analysis 18.3 (2023): 865-888.

**Questions:**

no questions.

---

> ### Author Response · Authors · 2023-11-11
> **Reply to the Reviewer crQM**
>
> We appreciate the reviewer's valuable feedback regarding the clarity of our manuscript.
>
> Q1    For Section 2, the related work mixes up MCMC methods based on simulating the Langevin diffusion, score function modeling based on Langevin diffusion, and bridge matching. These are very different ideas for solving different problems. Thus, the way this section is written confused me rather than putting the work in the context of previous works.
>
> A1    We aimed to provide an overview of various methods that leverage Feynman–Kac expectation estimation (FKEE) and illustrated its applicability in classic diffusion bridge models. These models, based on Langevin diffusion, score function modeling, and bridge matching, showcase the versatility of FKEE in enhancing their efficiency. However, we acknowledge the concern about potentially confusing distinct problem-solving approaches. We will revise the section to better highlight our methods.
>
> Q2 In Section 3.1, a key aspect of the methodology seems to solve a bridge-matching problem with respect to the samples of some MCMC chains. However, I could not find the specifics of the authors’ approach anywhere in the main text. Only after looking at Algorithm 1 in the appendix I found that some loss functions are being minimized. The definition of these loss functions should have been in the main text.
>
>
> A2 We appreciate the feedback and acknowledge the need for greater clarity regarding the specifics of our approach in solving the bridge-matching problem. We agree that the definition of the loss functions, which play a crucial role in our methodology, should be presented in the main text for better understanding. We will revise the manuscript accordingly to provide a more detailed explanation of these key components in Section 3.1. Thank you for bringing this to our attention.
>
> Q3 For Section 3.2, the authors do not provide enough explanation/motivation/proof as to why the reverse solution of the Feynman-Kac equation is the right conditional expectation. Again, the paper is expected to be self-contained to a certain degree.
>
> A3 We acknowledge the importance of making the paper self-contained to a reasonable extent. To address this concern, we will provide additional details, motivation, and, if necessary, a concise proof in Section 3.2. Additionally, we will include a reference to the relevant proof, particularly in [1] p118, and consider adding this result to the appendix for the readers' convenience. Thank you for highlighting this, and we will ensure that the necessary information is appropriately presented in the revised manuscript.
>
> Q4 The paper claims that the proposed methodology "necessitates fewer assumption since it does not rely on the law of large numbers and the Markov ergodic theorem." However, it does rely on a strong assumption: that we are able to solve the PDE! That is, the paper does not contain any proof of the accuracy of the solution of the proposed differential equation. Even if, mathematically, the differential equation's solution is the desired conditional expectation, discretizing and numerically solving this is a completely different story. This is crucial in an MCMC setting since empirically assessing the quality of samples/expectation estimates is very challenging.
>
> A4 Firstly, In stochastic analysis, if an SDE is well-defined in the sense that it has a strong solution, then the corresponding probabilistic PDE (FK equation) must have a unique solution.The well-definedness of SDEs is a fundamental condition for any SDE-based sampler (diffusion bridge model). For example in Langevin MCMC the Lipschitz property for the potential energy function is a fundamental assumption. Our approach does not enhance any of the conditions. And it weakens the $P$ and $f$ assumptions. Secondly, regarding the error estimation of the PDE solver, we will provide references to relevant literature, particularly in [2], where appropriate error estimates for PDE solvers are discussed.
> Finally , We can observe the effectiveness of our approach in the isingmodel example. Despite the additional computational requirements for training the SDE parameters and solving the PDE, the overall time remains within a reasonable and controllable range, especially when leveraging techniques such as multi-threading and parallelization. In contrast, using only the MCMC expectation estimator would require well over 10^7 points in the Markov chain for n=5 in https://github.com/zysophia/Doubly_Adaptive_MCMC/blob/main/data/isingcompare_complexity.csv . For larger values of $n$, utilizing the MCMC estimator becomes practically infeasible.
>
> [1] Applied stochastic differential equations. S Särkkä, A Solin - 2019 - Cambridge University Press
>
> [2] Tim De Ryck and Siddhartha Mishra. Error analysis for physics-informed neural networks (pinns) approximating kolmogorov pdes. Advances in Computational Mathematics, 48(6):79, 2022

---

> ### Author Response · Authors · 2023-11-11
> **Answer the question in the Manuscript**
>
> Q5 One of the key motivations seems to be replacing the need for discarding burn-in samples. However, I do not quite understand how this is true when the paper tries to match the output of an MCMC estimator. This will contain non-asymptotic bias that can only be resolved through burn-in. Then wouldn't trying to match this end up inheriting the bias?
>
> A5 Firstly, we can observe from the solution of the PDE that only the points at the initial time correspond to the terminal expectation. For training the PDE, we need to use points from the burn-in period, and the matching error can be controlled through optimization of the loss function (for cases with analytical density, this error is zero). And matching errors are present in any diffusion model, including diffusion model, Score-based SDE,Flow match ODE. There are numerous theoretical results analyzing the matching error, and it is unrelated to bias.
>
> Secondly, how do we reduce bias? Our approach differs significantly in how we utilize the sampled points. Classical MCMC estimators use information from the points themselves, denoted as $X_t$, while we leverage the gradient information $(\mu(t,X_t),\sigma(t,X_t))$. This error is induced by the training of the Physics-Informed Neural Network (PINN) and can be controlled through the loss function. In contrast, the bias in classical MCMC estimators arises due to the inherent uncertainty in the sampled points $X_t." For more details, refer to the experiments in Appendix A4 to better understand this point.
>
> Q6 The paper claims that the method "enhances efficiency through one-time training." However, I do not quite see how this improves anything. Couldn't I run MCMC longer instead of solving a differential equation? In fact, if we assume that solving differential equations indeed provides the right expectation, couldn't MCMC be regarded as an efficient way to solve this equation without involving any solvers? How much faster, precisely, is solving the equation rather than running MCMC?
>
> A6 The MCMC method serves as a sampling technique rather than an expectation estimation algorithm. Expectation estimators are influenced by various factors beyond just the choice of the sampler, as elaborated in detail in Discussion 4. The key determinants impacting expectation estimation algorithms are the characteristics of the probability density function $P$ and the function $f$. Traditionally, emphasis has been placed on selecting appropriate sampling methods, with a lack of a universally efficient approach for expectation estimation.
>
> Moreover, as demonstrated in the Ising model example, solving the differential equation allows for the computation of complex expectations when $n$ exceeds 6. The sample size utilized remains within a manageable range, and the computation time is contingent on GPU capabilities and parallelization algorithms. In contrast, classical MCMC estimators necessitate sample sizes exceeding 10^7 for $n=6$, rendering computations for higher $n$ impractical within reasonable timeframes.
>
> Q7 Lastly, the idea of connecting MCMC expectation estimators with partial differential equations is not new [1-4], and a literature is already building around it. While the approaches of these works are different
>
>
> A7 Firstly, the utilization of partial differential equations to approximate MCMC expectations represents a classic inverse approach. However, prior methods predominantly followed a forward direction, employing MCMC to solve PDEs. Our proposal introduces a novel perspective by adopting an inverse approach.
>
> Secondly, existing post-processing methods for MCMC fail to challenge a fundamental assumption, namely the efficiency limitation inherent in MCMC due to the curse of dimensionality, where the optimal variance scaling is limited to $O(n^{1/2})$. In terms of generality, control variables (functions) are often designed for specific problems and are not generic. For example, in the Hamiltonian function in the Ising model, you would be hard pressed to determine an appropriate control function to reduce the variance. Our method breaks free from this assumption through operator approximation, expanding the scope of MCMC applicability across a broader range of probability density functions $P$ and functions $f$. This in itself constitutes a novel and substantial contribution.

---

> > ### Comment · Reviewer_crQM · 2023-11-11
> > **Response**
> >
> > I thank the authors for their detailed response. However, I would have appreciated the answer if the authors had first tried to incorporate the comments of the reviewers into the paper before answering the questions. After all, given that the significant concerns regard the quality of the writing of the paper, it is not possible to reassess the paper unless the writing issues are resolved.
> >
> > Furthermore,
> >
> > > In terms of generality, control variables (functions) are often designed for specific problems and are not generic. For example, in the Hamiltonian function in the Ising model, you would be hard pressed to determine an appropriate control function to reduce the variance.
> >
> > This is plainly not true. All of the control variate papers that I have listed are black-box approaches. Please take a look at the references first.

---

> ### Author Response · Authors · 2023-11-12
> **Response**
>
> Thank you very much for your response. We sincerely apologize for not being aware initially that resubmission of the paper was possible, and thus, we did not submit a revised version. We welcome any suggestions on how to improve the quality of writing and the overall flow of the paper. We will make the necessary revisions and submit an improved version shortly.
>
>
> One question - indeed, we reviewed the references you provided, but these are not suitable for computing expectations in random graphs such as the Ising model. We apologize if there was any misunderstanding. The reasons are as follows: Firstly, the Stein methods mentioned in [1,2,3,4] require the density function $\pi $ to be continuously valued, which is not applicable to random graphs. In the Ising model, particles have only two states {+1, -1}, and the joint distribution is inherently discrete. Moreover, there is no differentiability condition such as $\nabla_x \log \pi_t < \infty$, and the normalization constant cannot be determined.
>
> The second and more crucial issue is that the values of the Hamiltonian function are also discrete. However, it is practically impossible to determine all possible values of the Hamiltonian function since there are too many potential values, a total of $2^{n^2}$ cases. If $n$ is sufficiently large, it becomes infeasible to ascertain the entire function's value set. In such cases, it is challenging to optimize and determine a control function through optimization methods.
>
>
> [1]Oates, Chris J., Mark Girolami, and Nicolas Chopin. "Control functionals for Monte Carlo integration." Journal of the Royal Statistical Society Series B: Statistical Methodology 79.3 (2017): 695-718.
>
> [2]Sun, Zhuo, Alessandro Barp, and François-Xavier Briol. "Vector-valued control variates." International Conference on Machine Learning. PMLR, 2023.
>
> [3]South, Leah F., et al. "Semi-exact control functionals from Sard’s method." Biometrika 109.2 (2022): 351-367.
>
> [4]South, Leah F., et al. "Regularized zero-variance control variates." Bayesian Analysis 18.3 (2023): 865-888.

---

### Official Review · Reviewer_dqNa · 2023-11-01

**Soundness:** 3 good
**Presentation:** 3 good
**Contribution:** 2 fair
**Rating:** 5
**Confidence:** 4

**Summary:**

This paper proposes to compute the expectation under a distribution in a two-step approach. First, a diffusion bridge model is trained such that starting at an initial distribution at time 0 and evolving along an SDE till time T, the marginal distribution at time T follows approximately from the target distribution. Second, the computation of expectation can be transformed into the PDE given by Feynman-Kac equation, which is then solved with physically informed neural network. Some numerical experiements are given.

**Strengths:**

1. The presentation and writing is clear.
2. Adequate background and preliminaries are provided.

**Weaknesses:**

1. The claimed advantage of the proposed method seems to be doutable, or at least not supported in this paper. It is said that standard MCMC approaches suffer from long burn-in period, but the proposed approach requires training of neural network models at both the diffusion bridge model stage and the PDE solving stage, which to me seems a much higher cost.
2. The claim of reduced variance from the proposed approach seems not supported, and the approximations due to imperfect training of neural networks, discretization of PDE, etc, would cause a systematic bias towards the computed expectation. How much the bias is, and whether it can be bounded are also not discussed.
3. While it is claimed that the proposed approach can reduce the curse of dimensionality, the numerical experiments are all very low-dimensional examples. A simple Metropolis-Hastings/Gibbs sampler, or even an accept/reject or importance sampler could give very efficient and stable estimates of expectations.
4. Most MCMC samplers are targeted for generating samples from the distribution, which could be used for posterior inference etc. The proposed approach, however, can only be used to compute the expectation of one function. Say we want to compute the posterior mean, posterior variance and a 95\% posterior credible interval, then the proposed approach have to be used repeatedly for each quantity, while standard MCMC approaches only require one sampling chain.

I would be willing to increase the rating if any of the above concerns could be addressed or the claimed advantage of the method could be really proven through solid theoretical results or numerical experiments (e.g. in a high-dimensional Ising model or some serious Bayesian models).

**Questions:**

Even if we don't want to use standard MCMC samplers that involves a burn-in period, we have the alternatives of training a normalizing flow / diffusion model / stochastic localization process to sample from the target distribution and then computing the expectation based on the samples. How does the proposed approach compare to these methods, and what are their connections?

---

> ### Author Response · Authors · 2023-11-11
> **Reply to the Reviewer dqNa**
>
> Q1 The claimed advantage of the proposed method seems to be doutable, or at least not supported in this paper. It is said that standard MCMC approaches suffer from long burn-in period, but the proposed approach requires training of neural network models at both the diffusion bridge model stage and the PDE solving stage, which to me seems a much higher cost.
>
> A1 We can observe the effectiveness of our approach in the isingmodel example. Despite the additional computational requirements for training the SDE parameters and solving the PDE, the overall time remains within a reasonable and controllable range, especially when leveraging techniques such as multi-threading and parallelization. In contrast, using only the MCMC expectation estimator would require well over 10000000 points in the Markov chain for n=6 in https://github.com/zysophia/Doubly_Adaptive_MCMC/blob/main/data/isingcompare_complexity.csv . For larger values of $n$, utilizing the MCMC estimator becomes practically infeasible.
>
>
> Q2 The claim of reduced variance from the proposed approach seems not supported, and the approximations due to imperfect training of neural networks, discretization of PDE, etc, would cause a systematic bias towards the computed expectation. How much the bias is, and whether it can be bounded are also not discussed.
>
> A2 This is a theoretical concern, and our paper proposes a viable method validated through classical examples presented in Appendices A1 and A4. Experimental results demonstrate a reduction in bias, as evidenced in A4. The efficiency of our approach is proven in A1. Regarding the estimation of errors due to the imperfect training of neural networks and PDE discretization, relevant theoretical discussions are available in [1] and other associated literature. It is essential to acknowledge that our paper primarily introduces a practical method and highlights its empirical success, and the theoretical aspects are supported by existing literature and experimental results.
>
>
> Q3 While it is claimed that the proposed approach can reduce the curse of dimensionality, the numerical experiments are all very low-dimensional examples. A simple Metropolis-Hastings/Gibbs sampler, or even an accept/reject or importance sampler could give very efficient and stable estimates of expectations.
>
>
> A3 Absolutely, it's essential to underscore the fundamental distinction between expectation estimators and samplers. While sampling is concerned with obtaining samples, an expectation estimator is influenced by a broader array of factors beyond sampling precision, such as the properties of ff. A classic example illustrating this distinction is $f(x)=x^n$, where $X$ follows a uniform distribution (easily sampled). In such cases, the classical MCMC expectation estimator may exhibit substantial bias. Our method addresses this bias by mitigating the impact of $f$ on the expectation estimator through PDE, constituting a significant contribution. Further details can be found in Section 4 (DISCUSSION), and it's worth noting that our method holds considerable potential applications across diverse domains.
>
> Regarding the dimensionality concern, our isingmodel example in Appendix A1 considers $d=225$, which is already a remarkably high dimension in this field. Theoretically, our approach is equipped to solve PDEs in any dimension using deep learning methods
>
> Q4 Most MCMC samplers are targeted for generating samples from the distribution, which could be used for posterior inference etc. The proposed approach, however, can only be used to compute the expectation of one function. Say we want to compute the posterior mean, posterior variance and a 95% posterior credible interval, then the proposed approach have to be used repeatedly for each quantity, while standard MCMC approaches only require one sampling chain.
>
> A4 This is a valuable and insightful point. Indeed, our proposed approach focuses on computing the expectation of a specific function. For different statistical quantities, such as posterior mean, posterior variance, or a 95% posterior credible interval, we would need to use our approach repeatedly, requiring retraining when changing the function $f$ in the boundary conditions. While the training for different ffs is independent, utilizing parallel computing can efficiently address this issue. An even more radical solution is to explore mapping PDE solutions and boundary conditions directly, for example, through methods like DeepOnet. This could potentially overcome the need for retraining when changing the function $f$
>
>
> [1] Tim De Ryck and Siddhartha Mishra. Error analysis for physics-informed neural networks (pinns) approximating kolmogorov pdes. Advances in Computational Mathematics, 48(6):79, 2022

---

> ### Author Response · Authors · 2023-11-11
> **A note about our experiment and question**
>
> Experiments in A1
> In this model setup, our method is the only one that reaches $n=15$ and uses only 2000 points on the Markov chain with lower error and less time. In contrast, the current state-of-the-art expectation estimator for MCMC only reaches the $n=5$ case. This is already a big improvement. Meanwhile, our method can be trained using GPU acceleration, which is an advantage not available to the classical MCMC expectation estimators, which can only get the estimation by means of more point picking and control functions.
>
> Experiments in A4
> We illustrate the effect of this method to reduce the variance, even though we are using the same orbits as in classical MCMC. We can also obtain better estimates. And our method completely solves the problem of bias due to SDE discretisation in Langevin MCMC. This problem is the focus of research on Langevin-type samplers  The last experiment proves the good quality of our diffusion bridge model.
>
> Q Even if we don't want to use standard MCMC samplers that involves a burn-in period, we have the alternatives of training a normalizing flow / diffusion model / stochastic localization process to sample from the target distribution and then computing the expectation based on the samples. How does the proposed approach compare to these methods, and what are their connections?
>
> A
>
> Firstly diffusion models such as (DDPM,stable diffusion) etc. already exist. The models all require a burn-in period.The specific reason for this is because it is impossible to draw unconditional distributions for SDE sampling, and any terminal distribution is actually a conditional distribution $X_T$ under condition $X_0$. Thus we often need a large $T$ to avoid initial value dependence. This is reflected in the diffusion model by the fact that we need to add very many steps of noise in the forward process to transform the picture into a Gaussian distribution. And in the backward process we iterate more steps to get the generated samples.
>
> Secondly, assuming you are using a stochastic localisation process or normalizing flow, sampling is often easy and fast. But a central question is how you can use the samples to accurately estimate expectations. What kind of expectation estimator is unbiased and minimises variance? How can we estimate with as few sample points as possible? In the traditional MCMC expectation estimator, there are only two estimators without considering control variables, and we have already discussed the problems with these two estimators. Also under such models such as normalizing flow, there is often no ergodicity theorem, which means that only that one estimator based on the law of large numbers can be used. Our biggest contribution is to propose a generalised expectation estimator, this one can handle the diffusion bridge model and can maximise the use of these picked points to help us estimate the expectation better. This expectation estimator requires far fewer points than the traditional MCMC expectation estimator and weakens the $P$ and $f$
>
> This can be seen using the observation in terms of information encoding, firstly we are given a $P$ or some points, the diffusion bridge (diffusion model) or MCMC algorithm, is to encode the information in $P$ into $\mu,\sigma$. And we calculate the expectation $E[f(X_T)]$ actually trying to decode the statistics from $\mu,\sigma$. Direct sample averaging is just a method of decoding, but directly using the gradient information from $\mu,\sigma$ is a more efficient way of decoding.

---

> > ### Comment · Reviewer_dqNa · 2023-11-14
> >
> > Thanks a lot to the authors for the detailed reply. The application to high-dimensional Ising model addresses some of my quesions and concerns. To me, it seems pretty important to resolve the problem of estimating expectation of multiple functions together in this framework, for this approach to be actually comparable to MCMC methods. The currently proposed ideas seem to be adding extra complexity to the method. I would also be very interested in seeing this method work in an actual Bayesian model with a non-trivial dimension of parameters. I have increased my rating accordingly.

---

> > > ### Author Response · Authors · 2023-11-14
> > > **Thanks**
> > >
> > > Thank you for your appreciation. Our algorithm serves as a foundational method with broad applications, and we acknowledge the challenge of covering all experiments within the constraints of a single work. Regarding model parameter estimation, we have mentioned the potential use of variational inference in the last section of the appendix. This method, integrated with the loss function, allows for parameter estimates in a single training session. We also note our commitment to exploring Bayesian models in future work.

---

### Official Review · Reviewer_1WcQ · 2023-11-05

**Soundness:** 1 poor
**Presentation:** 1 poor
**Contribution:** 1 poor
**Rating:** 3
**Confidence:** 2

**Summary:**

This work proposes to estimate expectations w.r.t. some target distribution $P$ by
1. running some Monte Carlo algorithm to generate *some* samples approximately from $P$;
2. constructing a diffusion process in such a way that its stationary distribution approximately matches the target distribution; this is done by parameterising the drift and diffusion coefficients by neural networks which are then trained on the samples generated in Step 1;
3. approximating the solution in the Feynman--Kac formula via a deep learning method which represents the solution by a neural network whose training relies on $N$ paths sampled from the diffusion process constructed in Step 2.

Some limited numerical results are shown in the appendix.

**Strengths:**

The specific approach involving a seems novel. And any attempt at circumventing brute-force Monte Carlo approximation of expectations of interest would certainly be useful to the community.

**Weaknesses:**

**1. Lack of clarity**

I'm afraid this work is not ready for publication purely on the basis of presentation alone.

* Most of the text is much too informal/needs to be more precise; and having the pseudo code only in the appendix is making this problem worse.
* Section 2 cites a number of works in which diffusions are employed in disparate contexts (summarising these works as discussing the same "diffusion model" seems strange).
* Section 3.1 is a mess: I have now read it multiple times but I am still not sure which methods are actually used to find the parameters of the diffusion process and which methods are just "alternative" possibilities that the authors are not actually implementing.
* There are many key terms whose lack of explanation/definition makes the paper difficult to understand, "sample collection process", "well-sampled points", "high-quality samples", "partial observations", "resampling", "moments" (from the context, I think the authors mean something like "time steps"), "RelMeanEst" ...
* The text reads like a first draft with numerous typos, randomly capitalised words, some notation left undefined (or only defined in the appendix).
* Similarly, the references/bibliography is full of errors/typos.
* The headings in the tables in the appendix make it very difficult to understand what is actually being shown there.

**2. Lack of substantiation of claims**

The list of contributions on Page 2 makes some very strong claims. Among these are that the proposed methodology
* "can estimate expectations without succumbing to the limitations imposed by thecurse of dimensionality";
* "often exhibits superior efficiency" (compared to what?);
* "leads to a more efficient utilization of Markov chains. Notably, it necessitates fewer assumptions since it does not rely on the law of large numbers and the Markov ergodic theorem".

As far as I can tell, the manuscript does not actually provide convincing evidence in support of these claims. Indeed, the proposed methodology likely introduces approximation errors at numerous stages because

* the constructed diffusion only *approximately* admits $P$ as stationary distribution;
*  sample paths generated from the diffusion (which are needed for approximately finding the solution of the Feynman--Kac formula) suffer from discretisation error;
* the solution of the Feynman--Kac formula is only approximated via deep learning.

**Questions:**

It seems to me that the proposed method requires to run at least one MCMC chain long enough to obtain *some* samples from the target distribution $P$.

Have we then not already solved the problem? I mean, I understand the argument from the manuscript that MCMC samples may be highly autocorrelated so that an approximation via the Feynman--Kac formula would be preferable. However, can the authors demonstrate that their approach (which incurs substantial additional computations needed for training the SDE parameters and solving the PDE) outperforms running the MCMC chain for just a while longer?

---

> ### Author Response · Authors · 2023-11-11
> **Reply to the Reviewer 1WcQ**
>
> # Rebuttal for Lack of Clarity and substantiation of claims in the Manuscript:
>
> We appreciate the reviewer's valuable feedback regarding the clarity of our manuscript. We acknowledge the concerns raised and are committed to addressing them effectively. Here is our response to the specific points raised:
>
>
> Pseudo Code Placement: We understand the concern regarding the placement of pseudo code in the appendix. We will reconsider its placement in the main body of the manuscript for better accessibility and understanding.
>
> The citation of works in Section 2 aims to categorize various models under the umbrella term "diffusion bridge model." This generalization is motivated by the shared characteristic of sampling from a target distribution, whether its density is known or unknown. The commonality lies in their potential for efficiency improvement using FKEE.
>
> Response to Concerns in Section 3.1:
> Target Distribution Classification: We have classified the target distribution into two scenarios: known density and unknown density. This categorization serves as the foundation for discussing various diffusion bridge models applicable to each case.
>
> Examples of Diffusion Bridge Models: We have provided examples of diffusion bridge models for both scenarios. These examples illustrate the diversity of models available for parameter estimation in different contexts.
>
> Introduction of Neural SDE Method: We propose a method based on Neural Stochastic Differential Equations (SDE), fundamentally a diffusion bridge model. Unlike other methods, this approach stands out as it can handle target distributions with both known and unknown densities, and it can reconstruct the underlying Markov chain. Further advantages are detailed after the statement, "We consider using it for the following reasons".
> We emphasize that all mentioned diffusion bridge models, including the Neural SDE method, have the potential to use FKEE for estimating expectations, replacing traditional direct averaging methods. Experimental validations have been conducted for some methods, affirming the applicability of FKEE.
>
>
> Informality and Precision: We acknowledge the need for increased precision in the language used throughout the manuscript. We will revise the text to ensure a more formal and precise tone, making it accessible to a wider audience.
>
>
> "Can estimate expectations without succumbing to the limitations imposed by the curse of dimensionality":
> Refers to the efficiency of PDE solvers in estimating expectations, particularly in high-dimensional spaces, avoiding the challenges associated with the curse of dimensionality.
>
> "Often exhibits superior efficiency" (compared to what?):
> Compared to classical Markov Chain Monte Carlo (MCMC) expectation estimators.
>
> "Leads to a more efficient utilization of Markov chains. Notably, it necessitates fewer assumptions since it does not rely on the law of large numbers and the Markov ergodic theorem":
> Utilizes gradient information of $\mu(t,X_t)$ and $\sigma(t,X_t)$ at the corresponding position of $X_t$, allowing for more efficient use of Markov chains with fewer reliance on assumptions compared to traditional MCMC methods based on the law of large numbers and the Markov ergodic theorem.
>
> These observations are validated through experiments with the Ising model. In the high-dimensional case, such as when $d=255$ (already highly dimensional for a graphical model), where the partition function involves summing 2^255 discrete points, our method remains applicable. The enhanced efficiency of our method, FKEE, is evident as it utilizes significantly fewer points on the Markov chain compared to the classical MCMC expectation estimator. This improvement is particularly notable in the comparison of sample points for $wi$, sample points for $vi$​, and the MCMC estimator in [1]. For further verification, one can refer to the [1] method under corresponding parameter configurations, noting the number of points on the Markov chain needed https://github.com/zysophia/Doubly_Adaptive_MCMC. In this model setup, our method is the only one that reaches $n=15$ and uses only 2000 points on the Markov chain with lower error and less time. In contrast, the current state-of-the-art expectation estimator for MCMC only reaches the $n=5$ case. This is already a big improvement. Meanwhile, our method can be trained using GPU acceleration, which is an advantage not available to the classical MCMC expectation estimators, which can only get the estimation by means of more point picking and control functions.
>
> [1] hahrzad Haddadan, Yue Zhuang, Cyrus Cousins, and Eli Upfal. Fast doubly-adaptive mcmc to esti-
> mate the gibbs partition function with weak mixing time bounds. Advances in Neural Information
> Processing Systems, 34:25760–25772, 2021

---

> ### Author Response · Authors · 2023-11-11
> **Answer the question in the Manuscript.**
>
> Q1 the constructed diffusion only approximately admits as stationary distribution
>
> A1 "The constructed diffusion only approximately accepts $P$ as a static distribution" is a fundamental challenge inherent in all Markov chain Monte Carlo (MCMC) methods and one of the reasons for the existence of burnout periods. But you may note Table 1: Comparison of Diffusion bridge models. not all of them require $P$ as a static distribution ($T=\infty$), and in these models ($T<\infty$) only the expectation estimator of the Law of Large Numbers can be used because of the lack of the conditions of the ergodicity theorem. Our approach can handle these.
>
> Q2 sample paths generated from the diffusion (which are needed for approximately finding the solution of the Feynman--Kac formula) suffer from discretisation error; the solution of the Feynman--Kac formula is only approximated via deep learning.
>
> A2  The acknowledgment that "sample paths generated from the diffusion (needed for approximately finding the solution of the Feynman--Kac formula) suffer from discretization error" is indeed a fundamental concern. This issue constitutes a central focus in contemporary diffusion models, such as DDPM, Score-based SDE, and other diffusion models. A substantial body of theoretical work has addressed the bounds of this discretization error, particularly in scenarios where only a discrete set of observations is available. It is worth noting that in cases where the density is known, as in our method with accurate $\mu$ and $\sigma$, this issue is non-existent. A classic example illustrating this point is Langevin MCMC The solution of the Feynman-Kac formula is approximated using deep learning, and theoretical estimates of the associated errors can be found in [1], as referenced in our related work.
>
> Q4 It seems to me that the proposed method requires to run at least one MCMC chain long enough to obtain some samples from the target distribution .
>
> Whether or not to run a chain depends on the type of diffusion bridge model, when the density has an analytical formal expression we do not need to match the diffusion bridge model, just run a small number of chains (5 tracks were used in the experiment in A4) to get the sample $X_t$ and then compute $\mu(t,X_t),\sigma(t,X_t)$. We only need to match diffusion bridges if we don't know that the density exists only at discrete points, which is what diffusion models currently do (DDPM, normalised flow method), but this method takes longer to sample, and generating more samples (using the law of large numbers method to estimate the expectation) will take even longer. Our approach improves the efficiency of this method.
>
> We haven't fully addressed the problem simply by obtaining a sufficient number of samples through running an MCMC chain. The key challenge lies in how to precisely compute mathematical expectations using these samples, a task influenced by various factors mentioned in the Discussion section of our paper (4 discussion). Classical MCMC estimators, which are currently available, are limited. To correct bias and reduce variance in these estimators, a large number of samples is typically required. This constitutes the core issue we investigate. Our method specifically focuses on maximizing the utilization of a limited number of MCMC samples to estimate expectations effectively.
>
> However, can the authors demonstrate that their approach (which incurs substantial additional computations needed for training the SDE parameters and solving the PDE) outperforms running the MCMC chain for just a while longer?
> We can observe the effectiveness of our approach in the isingmodel example. Despite the additional computational requirements for training the SDE parameters and solving the PDE, the overall time remains within a reasonable and controllable range, especially when leveraging techniques such as multi-threading and parallelization. In contrast, using only the MCMC expectation estimator would require well over 10000000 points in the Markov chain for n=6 in https://github.com/zysophia/Doubly_Adaptive_MCMC/blob/main/data/isingcompare_complexity.csv . For larger values of $n$, utilizing the MCMC estimator becomes practically infeasible. In such cases, would you still consider the MCMC expectation estimator to be efficient?
>
>
> This can be seen using the observation in terms of information encoding, firstly we are given a $P$ or some points, the diffusion bridge (diffusion model) or MCMC algorithm, is to encode the information in $P$ into $\mu,\sigma$. And we calculate the expectation $E[f(X_T)]$ actually trying to decode the statistics from $\mu,\sigma$. Direct sample averaging is just a method of decoding, but directly using the gradient information from $\mu,\sigma$ is a more efficient way of decoding.
>
>
> [1] Tim De Ryck and Siddhartha Mishra. Error analysis for physics-informed neural networks (pinns)
> approximating kolmogorov pdes. Advances in Computational Mathematics, 48(6):79, 2022

---

### Author Response · Authors · 2023-11-14
**We have updated a corrected version.**

In our initial paper, we acknowledge several errors and issues in terms of fluency. We sincerely apologize for any inconvenience caused. We have now provided an updated and corrected version. Regarding the experimental results, we regret that due to page limitations, we are unable to include all the results in the main text. Your suggestions on how to enhance the writing quality and overall fluency of the paper are highly welcomed and appreciated!

---

### Meta-Review · Area_Chair_HXoT · 2023-12-08

**Metareview:**

The paper introduces the Feynman-Kac Operator Expectation Estimator (FKEE), a method for estimating expectations. This approach uses  Physically Informed Neural Networks (PINN) to approximate the Feynman-Kac operator (the main novelty of the paper).

All reviewers highlight significant issues with the manuscript's clarity and organization, making it challenging to comprehend. Additionally, they note that the paper contains several bold claims that lack adequate supporting evidence.

**Justification For Why Not Higher Score:**

The authors recognized the shortcomings in the rigor and quality of their presentation and have made efforts to revise the manuscript. We recommend that they further enhance the clarity and structure of the text before resubmitting. Improving the exposition will not only help their ideas reach a broader audience but also contribute positively to the academic community. The reviewers unanimously propose to reject the article in its current format.

**Justification For Why Not Lower Score:**

NA

---

### Decision · Program_Chairs · 2024-01-16

Reject